

# Molecular composition of clouds: a comparison between samples collected at tropical (Réunion Island, France) and mid-north (puy de Dôme, France) latitudes.

Lucas Pailler[1], Laurent Deguillaume[1,2], Hélène Lavanant[3], Isabelle Schmitz[3], Marie Hubert[3], Edith Nicol[4], Mickaël Ribeiro[1], Jean-Marc Pichon[2], Mickaël Vaïtilingom[5], Pamela Dominutti[1,#], Frédéric Burnet[6], Pierre Tulet[7], Maud Leriche[1,8], Angelica Bianco[1,*]

[1] Laboratoire de Météorologie Physique, UMR 6016, CNRS, Université Clermont Auvergne, 63178 Aubière, France.
[2] Observatoire de Physique du Globe de Clermont-Ferrand, UMS 833, CNRS, Université Clermont Auvergne, 63178 Aubière, France.
[3] Univ Rouen Normandie, INSA Rouen Normandie, CNRS, Normandie Univ, COBRA UMR 6014, INC3M FR 3038, F-76000 Rouen, France
[4] Laboratoire de Chimie Moléculaire (LCM), CNRS UMR 9168, Ecole Polytechnique, Institut Polytechnique de Paris, route de Saclay, 91128 Palaiseau cedex, France
[5] Laboratoire de Recherche en Géosciences et Énergies (LaRGE), EA 4539, Université des Antilles, Pointre-à-Pitre, 97110, France
[6] Centre National de Recherches Météorologiques (CNRM), Université de Toulouse, Météo-France, CNRS, Toulouse, France
[7] Laboratoire d'Aérologie (LAERO), UMR 5560, CNRS, Université Toulouse III, IRD, Toulouse, 31400, France
[8] Centre pour l'étude et la simulation du climat à l'échelle régionale, Département des sciences de la terre et de l'atmosphère (ESCER), Université du Québec à Montréal, Montréal, H2X 3Y7, Canada
[#] now at Institut des Géosciences de l'Environnement (IGE), UMR 5001, CNRS, IRD, Université Grenoble Alpes, Grenoble, 38000, France

*Correspondence to*: Angelica Bianco (a.bianco@opgc.fr), Laurent Deguillaume (laurent.deguillaume@uca.fr)

**Abstract**

The composition of cloud water dissolved organic matter has been investigated through non-targeted high resolution mass spectrometry only on few samples, mostly collected in the Northern hemisphere, in USA, Europe and China. Nevertheless, there is a lack of measurements for clouds located in the Southern Hemisphere, under tropical conditions and influenced by forest emissions. Moreover, the comparison of the composition of cloud samples collected in different locations is not trivial, since the methodology for the analysis and data treatment are not standardized.

In this work, the chemical composition of three samples collected at Reunion Island (REU) during the BIO-MAÏDO field campaign, in the Indian Ocean, with influences from marine, anthropogenic and biogenic (tropical) emissions is investigated and compared to the chemical composition of samples collected at the puy de Dôme (PUY) observatory, in France. The same methodology of analysis and data treatment was employed, producing a unique dataset for the investigation of molecular composition of organic matter in cloud water. Besides the analysis of elemental composition, we investigated the carbon oxidation state (OSC) of dissolved organic matter, finding that overall samples collected at PUY are more oxidized than those collected at REU. Molecular formulas were also classified based on stoichiometric elemental ratios, showing the high frequency and abundance of reduced organic compounds, classified as lipids (LipidC), in this matrix, which led to search for terpenes oxidation products in cloud water samples.

To better discriminate between samples collected at PUY and at REU, statistical analysis (principal component analysis and agglomerative hierarchical clustering) was performed on the ensemble of molecular formulas and their intensities. Samples collected at REU, have a different composition from samples collected at PUY, mainly linked to the processing of organic matter in cloud water, but also to the influence of different primary emissions at the two locations.



## 1 Introduction

Clouds are an important missing piece in the atmospheric chemistry puzzle, affecting the composition of gas and aerosol phases. Soluble gases and soluble fraction of aerosol particles can dissolve in the cloud droplets, which represents an aqueous chemical reactor where chemical and biological transformations can occur. Transformations in cloud droplets consume chemical compounds and produce new ones, impacting their distribution among the various atmospheric phases (Herrmann et al., 2015). After cloud evaporation, these processes modify (1) the physicochemical properties of aerosol produced, such as oxidation state, chemical composition, and hygroscopicity (Isokääntä et al., 2022; Crumeyrolle et al., 2008) and (2) gas phase chemical composition. Cloud processes thus are known to play a role in air quality and climate.

In the past, the attention was focused on composition and concentration of the inorganic fraction of cloud water. In recent years, an increasing number of studies have approached the characterization of dissolved organic matter (DOM) and its processing by fogs and clouds. Fogs and clouds can contain up to $200\,\mathrm{mg\,C\,L^{-1}}$ depending on the environmental conditions (Herckes et al., 2013): the values are lower for clouds collected in remote and marine environments and tend to be higher for fogs, due to their proximity to the emission sources and their microphysical properties (smaller droplet size). The highest values were observed near urban areas and in clouds impacted by biomass burning (Ervens et al., 2013). The composition of DOM has been analysed in the past with targeted analytical approaches that enabled the quantification of certain classes of compounds, such as short-chain carboxylic acids, dicarboxylic acids, carbonyls and amino acids (Triesch et al., 2021; Dominutti et al., 2022; Zhao et al., 2019). These classes are selected as they are for example, soluble, markers of cloud chemical processing, tracers of the sources from the surface, or known to participate to the formation of aqueous secondary organic aerosol ("aqSOA"). However, their concentration accounts for only, on average 10-30% of the composition of DOM and an important fraction remains uncharacterized (Herckes et al., 2013). Non-targeted analysis using high-resolution mass spectrometry represents a new approach allowing to explore the composition of cloud water (Bianco et al., 2018; Cook et al., 2017; Mazzoleni et al., 2010; Zhao et al., 2013). It has already highlighted the complexity of the matrices, with contrasted level of chemical aging, with thousands of different detected compounds resulting from biogenic or anthropogenic sources and secondary products from atmospheric reactivity (Wozniak et al., 2008).

A large fraction of the works on cloud/fog chemical composition has been performed on samples collected in the Northern Hemisphere: a lack of measurements is observed in the Southern Hemisphere, especially at tropical latitudes, and/or in remote marine environments. These regions present highly different environmental conditions (temperature, sun irradiation) emissions and sources which could impact the distribution and transformations of the DOM in cloud water.

Réunion Island (REU), located in the Indian Ocean, is as a natural laboratory to analyse the photochemical and microbiological processes occurring in clouds forming under tropical conditions (Duflot et al., 2019). In the framework of the BIO-MAÏDO project, an intensive field campaign was conducted in 2019 in Saint Paul region of the Reunion Island. During this campaign, cloud samples were collected on the slope of the Mt. Maïdo where a state-of-the-art instrumentation was deployed to study the role of clouds on atmospheric chemistry (Leriche et al., 2023). In this frame, Dominutti et al. (Dominutti et al., 2022) reported the chemical composition and physical properties of 14 cloud events, collected in REU in March-April 2019. A highly comprehensive chemical screening has been performed in cloud water samples including ions, metals, oxidants, and organic matter (organic acids, sugars, amino acids, carbonyls, and volatile organic compounds). In particular, this work highlighted the high molecular complexity with elevated DOM content.

Cloud collection is performed routinely at the puy de Dôme (PUY) observatory, located in the Massif Central region of France at 1465 m a.s.l (above sea level). The PUY station is a remote sampling site allowing to collect air masses mostly influenced by marine emissions, transported over long distance. Physicochemical measurements are performed for each sample and reported in the free available database Puycloud (Deguillaume and Bianco, 2017) . Several cloud water samples collected at PUY have already been analysed by FT-ICR MS (Bianco et al., 2018, 2019a, b), but the analytical methodology adopted (analysis and data treatment) was slightly different from the one used in this study. Recently, we developed a new methodology



for the noise removal, the data assignment and the blank correction described in detail in (Pailler et al., 2022), which facilitate the comparison of the samples.

In this work, we present the analysis by high resolution mass spectrometry using FT-ICR MS (Fourier Transform – Ion Cyclotron Resonance Mass Spectrometry) of three cloud water samples collected at Reunion Island (REU) and of seven cloud water samples collected at puy de Dôme (PUY) in spring (2 samples), summer (1 sample) and autumn (4 samples) between

2019 and 2021 (Table 1). Only three samples collected at REU had enough volume to perform the FT-ICR MS analysis. For sampling at PUY and at REU, the same cloud water collectors were used (Vaïtilingom et al., 2024), two instruments with different magnetic fields (12 T and 9.4 T FT-ICR MS, from Bruker, with the same ionisation source) were used to perform analysis, and the same data treatment was applied to each sample. Our work provides a unique dataset for an improved comparison of the composition of cloud samples. The main goal of this study is to highlight the differences and similarities of

cloud water DOM collected in different environmental conditions but with the same analytical procedure.

## 2 Materials and methods

### 2.1 Sampling at REU

The BIO-MAÏDO field campaign was conducted between 14th March and 4th April 2019 in Réunion Island (REU) and is described in detail in Dominutti et al. (2022) and Leriche et al. (Leriche et al., 2023). Five sampling sites were instrumented

during the campaign along the slope of the Maïdo between 965 and 2165 m asl. During the campaign, clouds were forming quasi-daily along the slope at the end of the morning (El Gdachi et al., 2024).

Cloud sampling was performed at "Piste Omega" (21∘03′26″ S, 55∘22′05″ E) located at 1760 m a.s.l.. The sampling site is located in the western part of the island, along the road to the Maïdo peaks, around 5 km away from the Maïdo observatory (Baray et al., 2013). This site, corresponding to the forest between 1500 and 1900 m a.s.l., was chosen for the formation of

clouds on a daily basis. It is mainly surrounded by biogenic sources such as tropical forests characterized by the endemic tree species *Acacia heterophylla* (*Fabaceae*), locally called "Tamarinaie", and plantations of the coniferous species *Cryptomeria japonica* (*Taxodiaceae*). Below 1400 m a.s.l. the vegetation is characterized by scattered trees, and below 600-700 m a.s.l. by cultivation of sugarcanes. Cloud samples were obtained using a cloud collector (Deguillaume et al., 2014; Renard et al., 2020; Vaïtilingom et al., 2024) which collects cloud droplets larger than 7 µm (estimated cut-off diameter) by impaction onto

rectangular aluminium plates. The cloud collector was installed at the top of a 10 m mast, and cloud water samples were collected on an event basis. Before sampling, the cloud collector was rinsed thoroughly with Ultrapure-MilliQ water, and the aluminium plates, the funnel, and the container were autoclaved to avoid biological contamination. At the beginning of the campaign, sterilized MilliQ water was spread on the clean cloud collector and analysed with different approaches to detect any chemical or biological contamination. The samples were immediately filtered after collection on a 0.20 µm nylon filter to

eliminate microorganisms and micrometric particle residues. For three samples, named R8, R9 and R10B following the numbering of clouds proposed by Dominutti et al. (2022), the volume was large enough to perform FT-ICR MS analysis: aliquots of 50 mL were frozen in polypropylene vial and kept at -18°C until the analysis.

Back trajectory plots of the air masses were computed with Meso-CAT (Rocco et al., 2022) resulting from the coupling between high-resolution Meso-NH simulations and the lagrangian tool CAT (Computing Advection-interpolation of

atmospheric parameters and Trajectory tool (Baray et al., 2020)).

### 2.2 Sampling at PUY

The PUY observatory belongs to the atmospheric survey networks EMEP (European Monitoring and Evaluation Program), GAW (Global Atmosphere Watch), and ACTRIS (Aerosols, Clouds, and Trace gases Research Infrastructure); and it is part of the fully instrumented platform for atmospheric research Cezeaux-Opme-puy de Dôme (CO-PDD) (Baray et al., 2020).



Cloud water sampling was performed with the same cloud collector deployed at REU, but installed on the roof of the observatory (1465 m a.s.l.). The aluminum impactor was cleaned and sterilized by autoclaving before each cloud collection. Samples were collected in sterilized bottles and cloud water was filtered using a 0.20 μm nylon filter. pH was measured immediately after sampling. Oxidants concentration, concentration of main inorganic and organic ions, and dissolved organic carbon were measured on frozen aliquots. An aliquot of each sample was frozen on site and stored in appropriate vessels at

−20 °C until these samples were analysed by mass spectrometry (FT-ICR MS). More details about the physicochemical analysis are reported in (Bianco et al., 2015).

The air mass origins for each event were calculated using a 72-h back-trajectory obtained by using the CAT model (Baray et al., 2020) at an interception height of 1465 m a.s.l., corresponding to the PUY summit.

### 2.3 Sample treatment, FT-ICR MS analysis and data treatment

The cloud water samples were thawed at ambient temperature (≈20 °C) in a bench hood. Solid phase extraction (SPE) was used to concentrate the cloud DOM and remove inorganic salts before FT-ICR MS analysis. Strata-X (Phenomenex) cartridges (1 g of sorbent contained in TEFLON® tubes) were used for SPE following the procedure already described in (Pailler et al., 2022). SPE extracts were stocked at 4°C in brown glass vials with TEFLON® cap until analysis. The sample blank was extracted by SPE using the same procedure used for the cloud water sample.

High-resolution mass spectrometry analysis was performed using a FT-ICR SolarixXR 12T (Bruker, Germany) (Cobra Laboratory, INSA Rouen, France) for REU samples and a FT-ICR SolariX XR 9.4T (Bruker, Germany) (LCM Laboratory, Ecole Polytechnique de Massy Palaiseau, France) for PUY samples, both equipped with an electrospray ionization (ESI, Bruker) source, set in the negative ionization mode. The instruments with higher resolving powers are expected to assign a larger number of molecular formulas within the chosen margin of tolerance, as more isobaric species will be separated. Using

a similar tuning for 12 T and 9.4 T FT-ICR MS, the number of assigned molecular formulas were found to be different depending on the sample and resolving power of the instrument. Contrary to resolving power, ion suppression effects can exist in ESI to reduce the number of assigned MFs, as in samples 22/10/2019, 17/07/2020 and 03/11/2020. To atone for this variability in instrument response, both from effects of ionization and resolving power, we therefore chose to use a conservative approach by working with relative number of occurrence (or relative weighted occurrence) of molecular formulas within a

class and similar formula assignment routines (Hawkes et al., 2020). The instruments were externally calibrated with sodium trifluoroacetate (NaTFA). Samples were infused directly into the ESI source. The parameters were optimized to obtain a stable ion current with a minima ion injecting time into the mass analyser. The infusion flow rate was 2.0 μL min⁻¹, the drying gas temperature was 200 °C, and the drying and nebulizing gas flow rate were 4.0 L min⁻¹ and 1 bar, respectively. The ESI capillary voltage was 3.6 kV in negative ion mode. One hundred scans were accumulated for each spectrum. Methanol was injected

prior to the injection of each sample, and the acquisition was performed to evaluate the potential presence of residual pollutants. The acquisition size was set to 8 M, resulting in a mass resolving power of $(11.8 \pm 2.1) \times 10^5$ for REU samples at 200 $m/z$ and $(4.4 \pm 0.6) \times 10^5$ for REU samples at 600 $m/z$. PUY samples were acquired with a slightly lower resolution of $(7.5 \pm 0.3) \times 10^5$ at 200 $m/z$ and $(2.6 \pm 0.3) \times 10^5$ at 600 $m/z$.

Each spectrum was preliminarily treated after acquisition with Bruker Data Analysis. Spectra were internally recalibrated using

the recalibrant list reported in Pailler et al. (2022) and the peak-lists were extracted with signal to noise (S/N) ratio higher than 7. The noise was determined by considering the whole mass range of the spectrum. Formula assignment was performed with MFAssignR, using the procedure detailed in (Pailler et al., 2022). For clarity, recalibration and S/N correction were not applied to the peak-list since they were already performed with Data Analysis. Nevertheless, we decided to apply isotope filtering, which is a crucial step to avoid incorrect monoisotopic assignment. This feature allows a tentative filtering of masses with ¹³C

and ³⁴S from the mass list and creates a separate mass list containing the monoisotopic masses along with the polyisotopic masses and their types (¹³C or ³⁴S isotopes). Those lists were used as separate inputs in the MFAssignR molecular formula





(MF) assignment. For the MF assignment, we searched for even- number electron deprotonated molecules $[M - H]^-$ with a single negative charge in the $m/z$ range 100–1000 Da; no radicals were allowed. The Double Bond Equivalent (DBE) was set to be in the range 0–25 in accordance with Giannopoulos et al. (Giannopoulos et al., 2021) and Koch et al. (Koch et al., 2005),

and the elemental composition was in the range $C_{1-70}H_{2-140}O_{1-25}N_{0-4}S_{0-1}P_{0-1}$ (Bianco et al., 2018). Phosphorous assignment is tricky in mass spectrometry as it has one single stable isotope. In order to avoid incorrect assignment of P, only C, H, N, O and S atoms are considered in the first assignment. Then, the unassigned masses are used to perform a second assignment considering P. It is a more reliable method to assign MFs without excluding P atoms that are significantly present in the environmental samples (Rivas-Ubach et al., 2018).The $m/z$ tolerance for the assignment was set to 0.1 ppm, and the DOM-

NOM (dissolved organic matter-natural organic matter) rules were chosen for the attribution (Koch et al., 2005; Koch and Dittmar, 2006). The assignment relies on a *de novo* calculation (where *de novo* stands for the first of a series) based on the $m/z$ value, set at 300 Da, the matching tolerance, and the elemental range constraint. $CH_2$ and $H_2$ MF extensions were used to extrapolate from a *de novo* calculation to find target peaks related by these natural patterns. The output of the function gives a list of ambiguous (multiple MFs that have been assigned to the same peak) and unambiguous (MFs that have been assigned to

a unique mass) MFs. Only MFs in the unambiguous list were considered for further data comparison because the list of ambiguous MFs was empty in most cases. Seven criteria were applied to exclude formulas that do not occur abundantly in natural organic matter: DBE must be an integer value, $0.2 \leq H/C \leq 2.4$, $O/C \leq 1.5$, $N/C \leq 0.5$, $S/C \leq 0.2$, $2 \leq H \leq (2C + 2)$, and $O \leq (C + 2)$. In the case of the assignment of multiple isotopes, only the monoisotope was considered. The peak-list of the blank sample was assigned with the same procedure used for the sample. MFs obtained for the blank were excluded from the

samples, without considering the intensity of the peaks.

In order to classify MFs, the methodology proposed by (Rivas-Ubach et al., 2018) was used in this work. The authors proposed a multidimensional stoichiometric constraint classification (MSCC) based on C, H, N, O, P and S stoichiometric ratios in six main categories (hereafter RUCs): LipidC, PeptideC, Amino-sugarC, CarbohydrateC, NucleotideC, and Oxy-aromaticC (Rivas-Ubach et al., 2018). This classification was already adopted for samples collected at PUY and was described in (Renard

et al., 2022).

### 2.4 Statistical analysis

The analysis of variance, ANOVA test, used to determine differences between parameters measured for different samples, such as OSC, was performed using the ANOVA-OneWay package of OriginPro. Principal component analysis (PCA) and agglomerative hierarchical clustering (AHC) were performed using OriginPro on the matrix containing the intensity of each

MF for the samples collected at REU and at PUY. In this matrix, "MFs" represented the scores (9251 MF) and "samples" the loadings (10 in total, 3 for REU and 7 for PUY). For samples from REU and PUY, the intensity of each MF was normalized by the sum of the intensity of the assigned MF for each sample, adopting the procedure described in Gurganus (Gurganus et al., 2015). The PCA was performed using Pearson correlation. The two principal components covered 50.7% of the variance of the matrix. For AHC, the same matrix was analysed using Ward method, which analyses the variance of clusters and is the

most suitable method for quantitative variables. The number of clusters was defined (2).

### 3 Results and Discussion

#### 3.1 Characterization of cloud samples: chemical composition and air mass history

The concentrations of inorganic ions and the analysis of the back trajectory of the air masses give useful information on the anthropogenic and biogenic influence of the cloudy air. Clouds are collected and physico-chemical measurements are

performed at PUY since 2001. Using a multicomponent statistical analysis of the concentrations of main inorganic ions and pH, clouds are classified into four different categories: polluted, continental, marine and highly marine, as described in Deguillaume et al. (2014) and Renard et al. (2020). In parallel, this chemical classification is confronted to air mass history




assessed with the back-trajectory CAT model. In this work, based on their chemical compositions, all the studied PUY sample are classified as "marine". This class is mainly characterized by its low content of ions. Only sample 17/07/2020, with $NH_4^+$

concentration of 339.8 µmol $L^{-1}$, could present influences from continental sources. The analysis of the back-trajectory shows that the air mass corresponding to this sample has effectively a marine origin, but has travelled at low altitude (below 2km) on the continental North sector before reaching the PUY summit (Table S1). Similarly, sample 08/10/2021 presents low ionic content but high dissolved organic content (DOC) value: the back trajectory plot shows the intrusion of air masses of continental and marine origins in the free troposphere, with an up-drift of the air masses at altitude of 5 km, and a consequent

descent of the air mass before reaching PUY summit. The liquid water content (LWC) of the cloud samples presented in this work is comparable with values measured in marine stratus and stratocumulus clouds.

The correlation between ion content and air mass history has been fully highlighted for PUY situation. For samples collected at REU, this is much less obvious since the circulation of air masses reaching the sampling site is completely different and highly variable in time and space. The Maïdo region located to the Northwest of the island is conditioned by the lifting of the

trade winds over the relief of the island and the circumvention of the trade winds around the relief (El Gdachi et al., 2024). This low and medium altitude flow leads to a return flow generally located in the west and north-west sector. All these circulations lead almost daily to cloud formation on the slopes of Maïdo in the morning and beginning of the afternoon and evaporation of the clouds at the end of the daytime (e.g. (Duflot et al., 2019; Rocco et al., 2022; Leriche et al., 2023). During the BIO-MAÏDO campaign, 14 cloud samples were collected and characterized by physico-chemical analysis. Samples R8,

R9 and R10B are in the average of the samples collected during the campaign and present LWC values mainly below 0.1 g m$^{-3}$, similarly to the values reported for fog (Gonser et al., 2012; Mazoyer et al., 2022). The ionic content in the three samples is elevated, due to the contribution of sea salt to the cloud formation ($Na^+$: $220 \pm 35$ µmol $L^{-1}$; $Cl^-$: $196 \pm 25$ µmol $L^{-1}$). Nitrates shows concentrations of $148 \pm 36$ µmol $L^{-1}$, probably linked to local anthropogenic sources, mainly located in the coastal zone. Anthropogenic compounds are then transported by anabatic breezes during the day, as confirmed by the high concentration of

sulfate ($87 \pm 26$ µmol $L^{-1}$). The pH of the cloud samples does not vary a lot, with values ranging from 5.2 to 5.5. Values are reported in Table 1.

The average concentration of DOC is equal to $10.2 \pm 6.0$ mgC $L^{-1}$ that is much higher than values reported in marine environments (Gioda et al., 2011) or at the PUY station (Deguillaume et al., 2014). R10B shows a particularly high value of DOC of 17.0 mgC $L^{-1}$. As for inorganics, this indicates additional inputs of DOC other than sea-related ones.

Table 1 reports the physical and chemical characteristics of the cloud events, such as dates, sampling period, liquid water content (LWC), mean effective diameter (Deff), temperature, pH and concentrations of total organic carbon (TOC) and terpenes in water. Ancillary measurements extracted from Dominutti et al. (2022) of carboxylic acids and carbonyls are reported in Table S2.

Table 1: Date, sampling time and microphysical and physicochemical characteristics of samples presented in this work. Yellow headlines depicts samples collected at REU and blue headlines at PUY. NM: not measured.

| | R8 | R9 | R10B | 02/03/2019 | 15/03/2019 | 02/10/2019 | 22/10/2019 | 17/07/2020 | 03/11/2020 | 08/10/2021 |
|---|---|---|---|---|---|---|---|---|---|---|
| Date | 28/03/2019 | 30/03/2019 | 01/04/2019 | 02/03/2019 | 15/03/2019 | 02/10/2019 | 22/10/2019 | 17/07/2020 | 03/11/2020 | 08/10/2021 |
| Time (start-end UTC) | 07:15-12:31 | 07:54-14:30 | 11:18-14:30 | 8:20-9:30 | 7:30-9:30 | 12:35-14:45 | 12:40-14:30 | 9:36-12:40 | 7:40-12:40 | 7:40-10:10 |
| Season | Autumn | Autumn | Autumn | Winter | Winter | Autumn | Autumn | Summer | Autumn | Autumn |
| LWC [g m$^{-3}$] | 0.13 | 0.09 | 0.07 | NM | NM | NM | NM | 0.15 | 0.13 | 0.51 |
| Deff (µm) | 12.7 | 15.3 | 12.6 | NM | NM | NM | NM | 21.0 | 16.2 | 21.2 |



| Temperature (°C) | 17.5 | 17.2 | 16.6 | -0.2 | 3.0 | 6.5 | 6.1 | 10.0 | 2.0 | 2.0 |
|---|---|---|---|---|---|---|---|---|---|---|
| pH | 5.5 | 5.2 | 5.5 | 5.0 | NM | 5.2 | 5.8 | 5.4 | 5.1 | 4.5 |
| DOC [mgC L$^{-1}$] | 7.9 | 5.8 | 17.0 | 8.3 | NM | 6.8 | 11.2 | NM | 4.0 | 12.6 |
| Isoprene [ng mL$^{-1}$] | 2.8 | 2.1 | 3.3 | NM | NM | NM | NM | NM | NM | NM |
| α-pinene [ng mL$^{-1}$] | 0.1 | 9.7 | 0.3 | NM | NM | NM | NM | NM | NM | NM |
| β-pinene [ng mL$^{-1}$] | n.d. | 5.4 | 0.2 | NM | NM | NM | NM | NM | NM | NM |
| Limonene [ng mL$^{-1}$] | 0.1 | 5.2 | 0.3 | NM | NM | NM | NM | NM | NM | NM |
| SO$_4$$^{2-}$ (μmol L$^{-1}$) | 113.1 | 60.9 | 88.4 | 5.5 | 4.1 | 13.2 | 19.7 | 40.0 | 21.5 | 12.9 |
| NO$_3$$^{-}$ (μmol L$^{-1}$) | 163.4 | 107.8 | 173.9 | 12.4 | 9.3 | 42.6 | 0.1 | 162.1 | 12.8 | 31.8 |
| Cl$^{-}$ (μmol L$^{-1}$) | 190.0 | 175.0 | 224.0 | 36.3 | 24.3 | 109.0 | 35.1 | 39.6 | 37.8 | 11.6 |
| Na$^{+}$ (μmol L$^{-1}$) | 221.8 | 184.5 | 253.6 | 42.3 | 18.1 | 120.4 | 46.1 | 54.7 | 53.0 | 16.5 |
| NH$_4$$^{+}$ (μmol L$^{-1}$) | 126.4 | 81.1 | 130.7 | 13.1 | 6.4 | 4.2 | 0.0 | 339.8 | 61.8 | 33.5 |
| Mg$^{2+}$ (μmol L$^{-1}$) | 40.5 | 27.4 | 39.6 | 4.5 | 3.4 | 7.5 | 4.2 | 13.3 | 4.1 | 3.1 |
| K$^{+}$ (μmol L$^{-1}$) | 12.4 | 12.1 | 14.9 | 12.8 | 8.4 | 39.0 | 13.9 | 12.6 | 16.5 | 3.2 |
| Number of MFs | 3098 | 2503 | 2276 | 3244 | 2084 | 1543 | 120 | 715 | 312 | 7436 |

### 3.1 Molecular characterization

The numbers of assigned MFs for R8, R9 and R10B were 3098, 2503 and 2276, respectively, after blank exclusion (Table 1).

The MFs were grouped into the following groups, based on elemental composition: CHO, CHNO, CHOS, CHNOS, CHOP, CHNOP, CHOSP, CHNOSP, CHP, CHS, CHNP, CHSP, CHNSP and CHNS. Since the abundance of the groups CHP, CHS, CHNP, CHSP, CHNSP and CHNS was always below 0.2%, they will not be considered for the further analysis. The relative contribution of each group was calculated taking into account only the number of MFs (percentage in number) or the number and the intensity of the MFs (weighted percentage), as reported in Figure 1.

Considering the percentage in number, CHO accounted on average for 26.7±3.5%, with higher values for R9 and R10B (28.0 and 29.5 %, respectively) than R8 (22.8%). CHNO contribution was 45.5, 39.4 and 39.0% for R8, R9 and R10B respectively. Similarly, CHOS were more abundant in number in R9 and R10B (14.0 and 14.9%, respectively) than in R8 (11.6%), while CHNOS showed a different trend, with contribution in number of 8.1, 7.0 and 3.2% for R8, R9 and R10B, respectively. The most abundant group containing phosphorus in number of MFs was CHOSP, with abundance up to 9% for R8. The other

groups always contained less than 3% of MFs. Overall, the relative percentage of CHOP, CHNOP, CHOSP, CHNOSP, CHP, CHS, CHNP, CHSP, CHNSP and CHNS groups, in number of MFs, never overtake 14% and will be considered as "Others" in the following discussion.

The weighted percentages showed a slightly different trend: the CHO group was predominant for the three samples, with a weighted average percentage of 40.9±1.1%, while the CHNO group accounted for 28.0±3.8%, reaching 32.3% for R8. CHOS

and CHNOS groups accounted for 14.7±3.5 and 3.8±1.3%, respectively. In general, the contribution of each group was homogeneous for all the samples, with negligible differences between the groups. Interestingly, CHNO compounds presented





a higher number of compounds than CHO, but with a lower intensity: this result reflects the complex composition of the CHNO subgroup.

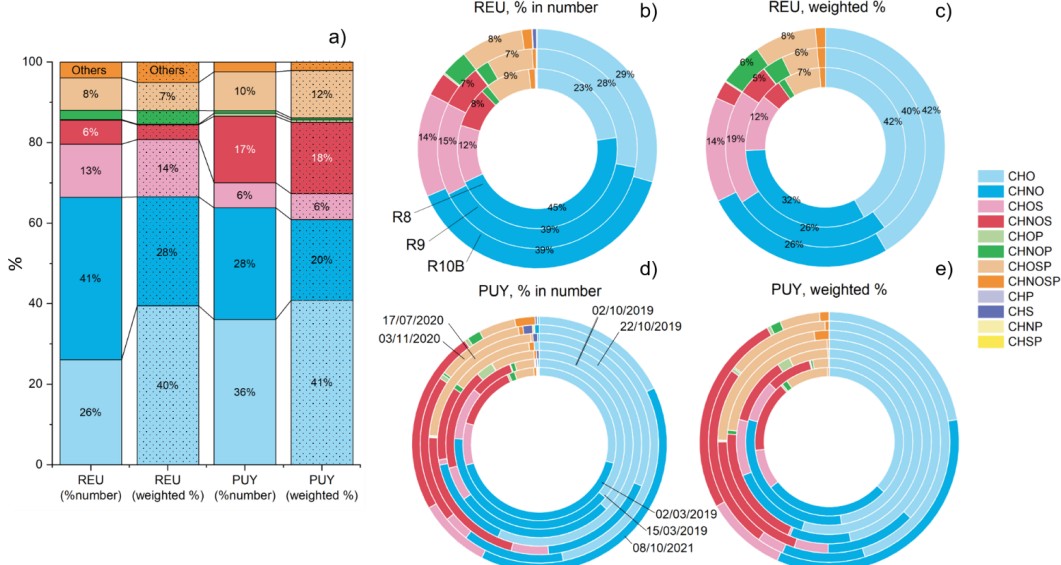

Figure 1: a) Average elemental composition in percentage of samples collected at REU and at PUY. Plain histograms represent the percentage on the number of MFs (% in number), while dotted histograms depict the percentage weighted for the intensity of the MFs (weighted %). "Others" category represents CHNOSP, CHP, CHS, CHNP and CHSP groups. b) and c) pie plots depict the elemental composition, in number of MFs and weighted for the intensity of the MFs, for R8 (inner circle), R9 (middle circle) and R10B (outer circle). Labels are reported only if the % is higher than 5%. d) and e) similarly to b) and c),
represent pie plots of the elemental composition of samples collected at PUY. The order of representation, from the inner to the outer circle, is 02/03/2019, 15/03/2019, 02/10/2019, 22/10/2019, 17/07/2020, 03/11/2020, and 08/10/2021. The legend is the same for all the plots.

The numbers of assigned MFs after blank exclusion for samples collected at PUY is highly variable, ranging from only 120
MFs for sample 22/10/2019 to 7436 MFs for sample 08/10/2021 (Table 1). The statistical analysis t-test was used to compare the averages of PUY and REU samples and determine if the differences are significant ($p<0.05$) or not ($p\geq0.05$). Interestingly, the PUY samples presented a lower abundance of CHNO compounds ($27.9\pm12.6$ percentage in number and $20.1\pm10.5$ weighted percentage), as reported in Figure 1. Consequently, the CHO contribution was $36.4\pm12.6$ in percentage in number, higher than the value observed for REU samples, while the weighted percentage, of $41.1\pm10.4$, was not different. Nevertheless,
these differences are not significant for CHO ($p\geq0.05$) and may be significant for CHNO ($p=0.058$). A larger dataset is needed to draw conclusions on this point. Conversely, the contribution of CHOS was significantly higher ($p<0.01$) in REU samples (13.5% in number, 14.7% in weight) than in PUY samples (6.3% in number, 6.5% in weight). One explanation is the possible influence of fresh marine emissions at REU that can be responsible for the emission of organo-sulfate compounds (Wang et al., 2023). Similarly, the CHNOS were significantly lower in samples collected at REU ($p<0.01$).
To go further, Figure S1 in the Supplementary Information reports the comparison of the number of carbons (#C), of hydrogen (#H), of oxygen (#O), of nitrogen (#N) and of sulphur (#S), the DBE, the elemental ratios (oxygen to carbon (O/C), hydrogen to carbon (H/C), nitrogen to carbon (N/C) and sulphur to carbon (S/C)), the carbon oxidation state (OSC), the aromaticity index (AI) and the CHO Index. The last three parameters are currently used in high resolution mass spectrometry to investigate, respectively, the oxidation state of organic matter (Dzepina et al., 2015), the presence of aliphatic and aromatic compounds



(Melendez-Perez et al., 2016) and the relative H, O, and C content in organic molecules (Mann et al., 2015). Figure 2a depicts the OSC for samples collected at REU (in yellow) and at PUY (in blue). OSC is a parameter that always increases with oxidation and was studied to investigate oxidation of DOM in aerosols, rains and fogs. Commonly, reported measurements by ESI of water-soluble organic carbon (WSOC) indicate OSC values lower than -0.9 for aerosol (in red in Figure 2a), between -0.9 and -0.7 for rainwater (in green) and higher than -0.7 for fog water (in purple) (Kroll et al., 2011). This increase in the OSC

suggests additional oxidation of WSOC in the aqueous phase. The ANOVA test, a statistical tool for the analysis of the difference between the means of more than two datasets, was used to investigate the differences of OSC between PUY and REU samples, as reported in Figure S2. R8 and R9 are similar whereas they are significantly different from R10B. Nevertheless, the three samples collected at REU are not significantly different from autumn samples collected at PUY (22/10/2019, 03/11/2020, 08/10/2021), with the exception of 02/10/2019. PUY dataset presents more heterogeneous values

than REU sample set, with a clear seasonal variation: winter samples have a lower OSC than summer and autumn samples, respectively of -0.97±0.56 (winter), -0.75±0.65 (autumn) and -0.60±0.58 (summer). This can be explained by the lower photoreactivity and the lower temperature during winter that decreases the efficiency of oxidation processes. The measured OSC for PUY autumn samples is not significantly different from the one measured at REU -0.77±0.55 ($p>0.05$); nevertheless, the environmental conditions are not comparable. This similarity could be due on one side to the presence of compounds

freshly emitted in REU samples that start to be processed in cloud water, with high temperature and oxidant concentration, and, on the other side, of moderately aged DOM from atmospheric processes at lower temperature at PUY. Temperature plays a key role in the processing of DOM: this is confirmed by the fact that samples collected at PUY during winter, with temperature below 3°C, have a lower OSC. For each season and location, the OSC values correspond to those observed for rainwater and fog water, which contain more oxidized DOM than aerosols due to aqueous phase transformations. This trend

is confirmed for all the subgroups but the values vary: for CHO compounds, OSC is -0.48±0.17 at PUY and -0.56±0.08 at REU, while for CHNO, it is -0.91±0.26 at PUY and -0.98±0.08 at REU, attesting that CHO compounds are generally more oxidized than CHNO. CHOS compounds, as reported in Figure S3, show an OSC value around -0.66 at PUY and -0.49 at REU, while CHNOS have OSC of -1.35 at PUY and -0.91 at REU, confirming the trend observed for CHNO.

Based on the study of Kroll et al. (Kroll et al., 2011), the plot reporting OSC as a function of the number of carbon atom ($n_C$),

presented in Figure 2b and 2c, gives additional insights into DOM composition and in atmospheric chemical processes such as oligomerization, functionalization and fragmentation of molecules. For more clarity, only the four autumn samples collected at PUY are reported in Figure 2c; winter and summer samples are reported in Figure S4. Based on the same study from Kroll et al. (2011), four classes were defined, accordingly to the $n_C$ and OSC values reported in Table S3. Hydrocarbon-like organic aerosol (HOA, dark red circles) and biomass burning organic aerosol (BBOA, red circles) correspond to primary particulate

matter directly emitted into the atmosphere. Semivolatile and low-volatility oxidized organic aerosol (SV-OOA and LV-OOA, orange and yellow squares, respectively) correspond to 'fresh' and 'aged' secondary aerosol produced by multistep oxidation reactions. The difference between samples collected at REU and at PUY is blatant: the dispersion of the formula in the OSC - $n_C$ space is similar, but the highest abundance for samples collected at REU is in the lower part of the diagram. DOM in samples collected at REU is less oxidized and comparable to HOA and BBOA, linked to primary emissions. The contribution

to each class can be evaluated through the boxplot reported in Figure 2d, calculated using the 7 samples from PUY and the 3 from REU. It shows that PUY samples have a lower fraction of HOA and a higher contribution of SV-OOA and LV-OOA than REU samples. This result is in line with those previously presented regarding OSC for CHO compounds. The lower right part of the OSC - $n_C$ space, corresponding to compounds with OSC below 1.5 and less than 10 carbon atoms is different for PUY and REU samples: compounds in this region can be recalcitrant to oxidation in cloud water. These results clearly assess

that DOM in REU samples is less oxidized than DOM in PUY samples.





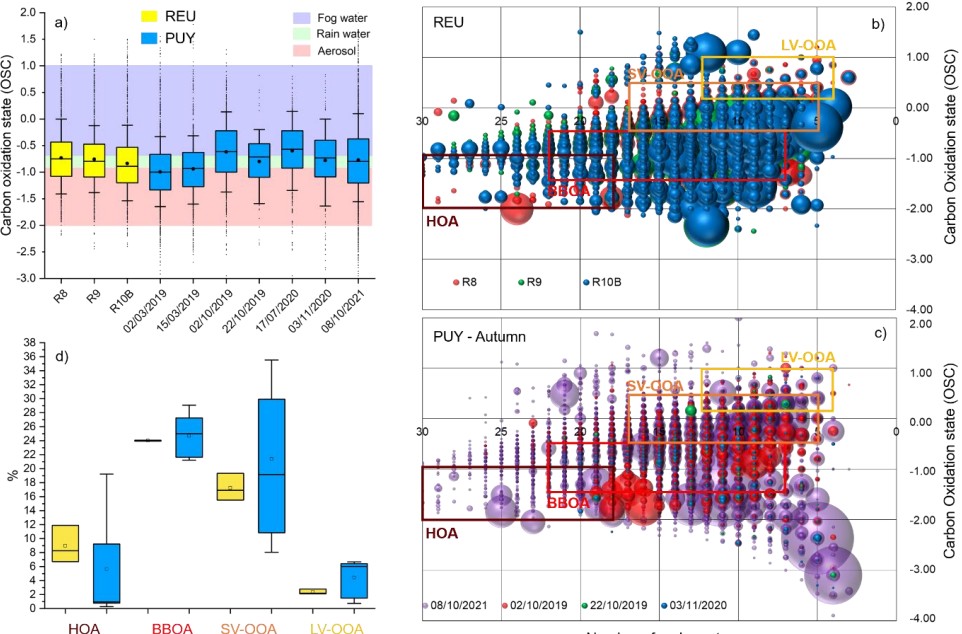

Figure 2: a) boxplot of the carbon oxidation state (OSC) for samples collected at REU (yellow) and at PUY (blue). The box represents the 1st-3rd quartile, the continuous line is the median, the dot is the mean and the whiskers the 10th and 90th percentiles. Outliers are depicted with dots. OSC ranges for aerosol, rain water and fog water are in red, green and purple, respectively (Kroll et al., 2011). b) bubble plot of the OSC – $n_C$ space for samples collected at REU. The intensity of the bubble is proportional to the intensity of the mass signal for all the formulas with the same OSC. Dark red square indicates hydrocarbon-like organic aerosol (HOA), red square biomass burning organic aerosol (BBOA), orange square semivolatile oxidized organic aerosol (SV-OOA) and yellow square low-volatility oxidized organic aerosol (LV-OOA). c) similarly to Figure 2b, bubble plot of the OSC – $n_C$ space for samples collected at PUY. d) boxplot of the abundance (in percentage) of the HOA, BBOA, SV-OOA and LV-OOA for samples collected at REU (yellow) and at PUY (blue). The box represents the 1st-3rd quartile, the continuous line is the median, the square is the mean and the whiskers the 10th and 90th percentiles.

## 3.2 Classification of molecular formulas on stoichiometric ratios – Rivas-Ubach classification

The MFs are difficult to associate with specific molecular structures, as each represents many possible isomeric arrangements, but rather allows categorizing molecules according to elemental composition, like CHO or CHNO. The elemental composition is usually represented graphically using van Krevelen diagrams (H/C vs. O/C). This graphical representation enables the study of the chemical composition of a sample, the visualization of structural classes of organic compounds (e.g., lipid, protein) (Kew et al., 2017; Kim et al., 2003) and the description of chemical reactions in terms of additions and losses in elementary compound categories (Heald et al., 2010). Nevertheless, it does not take into account the presence of heteroatoms, such as nitrogen, sulphur and phosphorus, leading to a significant overlap between the categories (Brockman et al., 2018; Rivas-Ubach et al., 2018). In our previous work (Renard et al., 2022), we used the multidimensional classification proposed by Rivas-Ubach et al. into six main categories: LipidC, PeptideC, Amino-sugarC, CarbohydrateC, NucleotideC and Oxy-aromaticC (Rivas-Ubach et al., 2018). The molecular composition of cloud water samples collected at REU and at PUY was investigated using the MSCC which is presented, for sample R8 in Figure 3a. Relative contributions in number of MFs (Figure 3b) are slightly different from relative contributions weighted by the intensity of the mass signal (Figure S5).



Regarding REU samples, the weighted proportion of the classes is homogeneous between samples, with 50-58% attributed to LipidC, 18-22% to Oxy-aromaticC, 4-8% to PeptideC and less than 5% to CarbohydrateC and Amino-sugarC. Up to 10% of the MFs were not assigned to any class (Not-matched), and no MF was assigned to NucleotideC (Figure S5). In MF number percentage, LipidC account for 50.3±5.8, Oxy-aromaticC for 24.6±2.4, PeptideC for 9.9±2.7, CarbohydrateC for 1.7±0.2 and Amino-sugarC for 4.0±1.1. MFs corresponding to Lipids are generally more intense than MFs in the other categories. The

same MF number distribution was observed for samples collected at PUY: 40.2±14.1% to LipidC, 22.6±6.9% to Oxy-aromaticC, 18.9±5.9% to PeptideC and less than 5% to CarbohydrateC and Amino-sugarC. Up to 12% of the MFs did not match to any class. The weighted values are similar, as reported in Figure 3b. Interestingly, the Rivas-Ubach classification highlights that samples collected at REU and at PUY are similar in terms of relative distribution between classes, with the only exception of PeptideC ($p<0.05$).

Nevertheless, the question on the high frequency and abundance of LipidC raises many questions, especially because (Renard et al., 2022) found almost the same abundance of LipidC for clouds collected under different environmental conditions (PUY) and analysed using a different FT-ICR MS ionization source (i.e., atmospheric pressure photoionization (APPI)).

LipidC contains a broad variety of organic compounds, from long chain carboxylic acids to sterols. Intuitively, there is more chance to detect long chains carboxylic acids than sterols in cloud water, but it is difficult to set-up a more systematic approach

to investigate LipidC category only studying MFs. To discriminate lipids into categories, we used the LIPID MAPS® database. This lipid classification system comprises eight lipid categories: fatty acyls, glycerolipids, glycerophospholypids, sphingolipids, sterol lipids, prenol lipids, saccharolipids and polyketides (Liebisch et al., 2020). The category "fatty acyls" contains some subcategories, such as fatty acids and conjugates, fatty alcohols and fatty aldehydes, which can be present in cloud samples since they are produced from the atmospheric oxidation of organic matter, and oxygenated hydrocarbons, mainly

linked to the oxidation of anthropogenic compounds (Wang et al., 2020). The category prenol lipids contains a subcategory "isoprenoids", which can be issued from oxidation of terpenes.

The MFs corresponding to the compounds listed in these subcategories were searched in the samples collected at REU and at PUY. In total 1071 compounds were selected: however, many of the selected compounds have different structural formula but identical MF; thus, a lower number of MFs was searched in PUY and REU samples. This research resulted in 500 MFs with

at least one occurrence in PUY or REU samples: 194 MFs corresponding uniquely to fatty acids, 11 to fatty alcohols, 3 to fatty aldehydes, 3 to oxygenated hydrocarbons, 124 to prenol lipids and 165 were in common between these categories. Working with MFs largely limited this approach, since most of the MFs of fatty acids were also contained in fatty aldehydes, fatty alcohols and oxygenated hydrocarbons, due to the different chemical function with the same type and number of atoms. However, most of the MFs attributed to prenol lipids were not shared with the other classes. This approach enabled to calculate

that fatty acyls, grouping fatty acids, alcohols, aldehydes and oxygenated hydrocarbons, contribute for PUY and REU samples from 0.6 to 8.3% in weight of the total detected ion intensity, and constitute a significant fraction of LipidC (from 11 to 46% in weight). Samples collected at REU have an occurrence of MFs attributed uniquely to fatty acids (12.7%) significantly higher than PUY samples (6.7%) ($p<0.05$). This is probably due to the contribution of sea salt, which is rich of surface active molecules, also called acid-soap complexes (Milsom et al., 2021). The contribution of the other categories is not significantly

different between REU and PUY. Isoprenoids represent from 1.3 to 4.0% in weight of the total abundance. One important outcome of this study is that a significant fraction of LipidC category (25.8±10.2, in weight) is composed of fatty acids, alcohols, aldehydes, oxygenated hydrocarbons and prenol lipids.



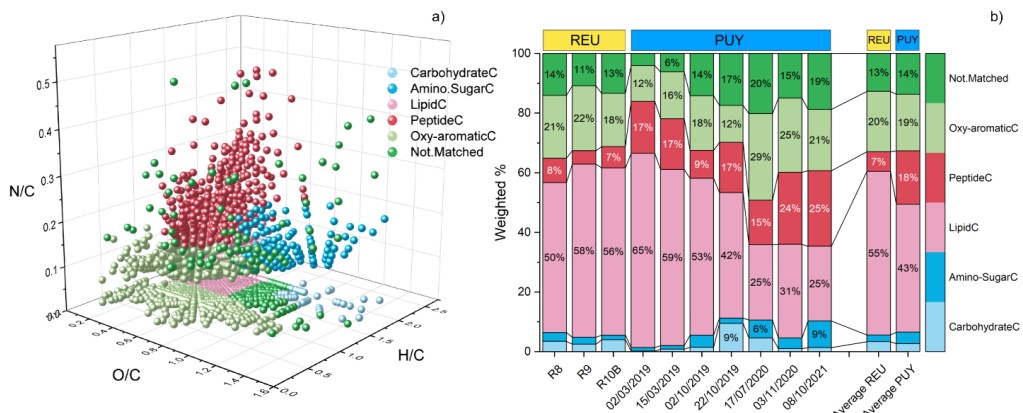

Figure 3: a) 3D representation of the Rivas-Ubach classification for sample R8. O/C ratio is reported on x-axis, H/C ratio on y-axis and N/C ratio on z-axis. Dots represent MFs, and the classification in Rivas-Ubach categories is depicted with the color-code reported in the legend. b) Stacked histogram of the relative contribution (in %) weighted for the intensity of the Rivas-Ubach categories for samples collected at REU and PUY.

### 3.3 Secondary organic aerosol tracers

An approach analogous to the one reported in Bianco et al. (Bianco et al., 2019a) was additionally used in this study to detect secondary organic aerosol (SOA) tracers, produced through volatile organic compounds (VOC) oxidation. MFs and their intensity for samples collected at PUY and at REU are reported in Table S4. Although the list of SOA is not exhaustive, it includes oxidation products from isoprene (Surratt et al., 2008; Riva et al., 2016a, b), alpha and beta-pinene (Surratt et al., 2008; Kourtchev et al., 2013; Kristensen et al., 2016; Finessi et al., 2014), limonene (Surratt et al., 2008; Kourtchev et al.,

2014), alpha and gamma terpinene and terpinolene (Surratt et al., 2008) and beta caryophyllene (Chan et al., 2011). In addition, we searched for MFs corresponding to nitroaromatic compounds (Zhang et al., 2013) and brown carbon (Lin et al., 2018; Budisulistiorini et al., 2017). The table also reports MFs corresponding to amino acids and sugars, particularly interesting for the evaluation of the presence of biogenic DOM.

The products of isoprene ozonolysis in the presence of sulfate (compounds $C_5H_8O_7S$, $C_8H_{14}O_{10}S$, $C_{10}H_{20}O_{10}S$, $C_5H_8O_5S$,

$C_5H_{10}O_5S$, $C_5H_{10}O_6S$, $C_8H_{10}O_4S$, $C_6H_{12}O_7S$, $C_9H_{14}O_6S$, $C_9H_{16}O_7S$ and $C_{10}H_{20}O_9S$) are present with very low occurrence both in PUY and REU samples. Similarly, isoprene epoxydiols (IEPOX) present few occurrences, with low signal at the two sites. $C_{10}H_{18}O_7S$, $C_{10}H_{18}O_8S$ and $C_{10}H_{20}O_8S$ are present in REU and PUY samples with low intensity, except for 08/10/2021 that shows a higher intensity for these MFs. Regarding other organosulfates produced from isoprene, only few MFs were observed in all the cloud water samples (REU, PUY). Lack of observation for organosulfates can be explained differently for PUY and

REU: at PUY, we made the hypothesis that organosulfates undergo hydrolysis in the aqueous phase, even at neutral pH (Bianco et al., 2019a). This hypothesis is supported by the low concentrations of isoprene in cloud water ($\approx$5-8 nmol L$^{-1}$, (Wang et al., 2020)). At REU, the concentration of isoprene is higher (40-50 nmol L$^{-1}$, (Dominutti et al., 2022)): it is emitted locally from vegetation, and in particular from the endemic species *Acacia heterophylla* (Tamarin), and its lifetime of 1.5h (Karl et al., 2004) may suggest that organosulfates are not produced yet by gaseous reactivity. However, there is a high probability of the

formation of organosulfates, due to the high concentrations of precursors (isoprene and sulphates). Interestingly, even if the air masses of the three samples collected at REU are quite different, the concentration of isoprene is similar and quite high: this result can be explained by the sampling just above the canopy of *Acacia heterophylla* that emits mainly isoprene (Rocco et al., 2024).



The concentration of alpha and beta pinene in cloud water is always higher than the one predicted by Henry's law (Wang et al., 2020; Dominutti et al., 2022; van Pinxteren et al., 2016). However, it was always very low in samples collected at PUY for alpha pinene (0.07-0.7 nM), while it varied over one order of magnitude for samples collected at REU. In particular, alpha pinene concentrations were 0.5, 71.5 and 2.0 nmol L$^{-1}$ and beta pinene concentrations were below detection limit, 39.9 and 1.3 nmol L$^{-1}$ for R8, R9 and R10B, respectively. This difference between REU samples can be explained analyzing the back trajectory plots of the air masses before arriving at the sampling site, as reported in Table S2. The air mass corresponding to R8 travelled at an altitude of 2.5-3.0 km a.s.l. and was surely less impacted by surface emissions than air masses corresponding to R9 and R10B, which travelled closer to the ground surface. The same trend was observed for other terpenes. Oxidation of alpha and beta pinene produces several identical MFs, corresponding to different chemical structures, which cannot be discerned by FT-ICR MS. Although different concentrations were detected in cloud water samples at REU, a similar number of MFs corresponding to organosulfates was detected in R8, R9 and R10B, with overall low intensity, which can be explained by the proximity of the sources of emission. Alpha and beta pinene oxidation products are present in samples collected at REU and at PUY during winter, while they were not found in summer and autumn samples at PUY, except 08/10/2021. As evidenced by the back-trajectory plots, summer and autumn samples at PUY undergo a long-range transport in the free troposphere that could be responsible of the degradation of organosulfates. Alpha pinene ozonolysis leads also to the formation of high weighted molecular compounds such as alpha pinene dimer esters (Kristensen et al., 2016), which were detected with high signal in samples collected at PUY. Few MFs were detected in REU samples, with low signal and comparable frequency between the three samples. Once more, the lower occurrence and the low signal of oxidation products in samples collected at the REU may be explained by the proximity of the biogenic sources, since the clouds were collected on a ten-meter mast in the slope of Maïdo Mountain, just above the canopy. The forest near to the collection site was mainly composed of the conifer *Cryptomeria japonica* (Dominutti et al., 2022). This would also explain the higher concentration of alpha and beta pinene in comparison to PUY samples. (Kourtchev et al., 2013) identified two MFs produced by alpha pinene oxidation in aerosol samples: $C_8H_{12}O_5$ (2-hydroxyterpenylic acid), which is one of the first oxidation products of alpha pinene, is detected in REU samples, with high intensity ($10^8$), but not in PUY samples, except in 02/10/2019. In contrast, $C_8H_{12}O_6$, 3-methyl-1,2,3-butanetricarboxylic acid (MBTCA), which is a product of the further oxidation of 2-hydroxyterpenylic acid, is not present in REU samples, confirming that the reactivity of alpha pinene in the atmosphere is still at its beginning. MBTCA is not present in PUY samples, except for 02/10/2019, where it is found with high intensity.

Similarly to other terpenes, limonene concentrations in cloud water has been shown to be 2-3 orders of magnitude higher than the values predicted by Henry's law constant, resulting into concentrations of 0.5, 38.5 and 2.3 ng L$^{-1}$ in R8, R9 and R10B (Dominutti et al., 2022). We observed that the MFs $C_9H_{14}O_4$, identified as an oxidation product from limonene of beta carophyllene, was present with high intensity in samples collected at REU and at PUY on 02/10/2019 (intensity of $10^8$), with similar intensity and trend than 2-hydroxyterpenylic acid. Organosulphates from limonene oxidation showed higher occurrence in samples collected at REU than at PUY, mainly due to the higher emission of limonene in REU compared to PUY. Nevertheless, the intensity of the FT-ICR MS signal was very low, apart from PUY sample 08/10/2021. Oxidation products of other terpenes, such as alpha-terpinene, gamma-terpinene and terpinolene, showed similar occurrence in samples collected at REU and PUY.

We also investigated the presence of beta caryophyllene derivatives because this molecule is specifically emitted by endogenous plant species of Reunion Island. The presence of N and S oxidation products of beta carophyllene could be explained by the fact that this terpene is emitted by plants located on the coast of the island, together with anthropogenic emissions and is afterwards oxidized during its transport along the slope of the island. Significant emission of this compound has been reported by Rocco et al. (Rocco et al., 2024). Beta caryophyllene could undergo nitration and oxidation in the presence of sulphates that lead to N and S containing compounds, leading to SOA (Surratt et al., 2008). Oxidation products in CHO subgroup were not detected in REU samples but were identified with high intensity at PUY samples. Conversely, oxidation



products containing N and S atoms were identified in REU samples with higher occurrence than PUY, even if the intensity of the signal was always very low.

Similarly to what is observed in PUY, samples collected at REU present low abundance of MFs corresponding to nitroaromatic
compounds and brown carbon.

### 3.4 Statistical analysis

The previous paragraphs highlighted the complexity of the DOM in samples collected at REU Island and at PUY, enabling to highlight differences and similarities on the samples for specific chemical parameters. However, it is difficult to find which of these parameters is the most relevant to point out the differences between cloud samples collected at the two sampling sites.
Although some contrasted characteristics can be found through the description of the molecular composition of clouds, we cannot conclude that cloud water DOM is different. To go beyond these limits, we used statistical tools to evaluate the differences between cloud water samples. We performed principal component analysis (PCA) and agglomerative hierarchical clustering (AHC) to explore similarities and hidden patterns among samples. The matrix is composed of 9251 scores (molecular formulas) and 10 loadings (3 samples from REU and 7 for PUY). Unfortunately, the small dataset does not enable
to find groups of MF typical of PUY or of REU: a larger number of samples is needed for this analysis.

We chose the representation of the results in a 2-D plot, although two principal components (PCs) are not enough to explain more than 48.7% of the variance of the dataset. In fact, to have a complete description of the variance (>85%), we need to extract at least 6 PCs (86.2%). The table of loadings, in Table S5, shows the correlation between the first six PCs and the samples (loadings). Samples collected at REU are mainly correlated with PC1, with values higher than 0.5, uncorrelated with
PC2 (-0.2) and mostly not correlated with the other PCs. Samples collected at PUY are more heterogeneous: all the samples, except 02/10/2019 and 02/03/2019, are correlated with PC2, with values higher than 0.25 but are also correlated significantly with the other PCs. PC3 could be identified as a "winter PC", since it is correlated with samples 02/03/2019 and 15/03/2019 and with no others, while it is more difficult discriminate between summer and autumn samples collected at PUY. However, the color-code of the table highlights that clouds collected at REU are statistically different than those collected at PUY, and
indicates a similarity of the DOM of samples collected at PUY that goes beyond the air mass origin and the seasonality . Figure S6 shows the contribution of the different PCs to the representation of each sample: the difference between REU samples, samples collected at PUY in winter and other samples collected at PUY is clearly visible.

The bi-plot, presented in Figure 4a, shows a good correlation between the samples collected at the same location, confirming that DOM for samples collected at REU is different from the DOM in cloud water collected at PUY. Sample corresponding to
02/10/2019 is not well represented in the 2-D plot and would have benefit of a 5-dimensional representation, since it is more represented by the fourth and fifth PCs (0.64 and 0.61, respectively, for PC4 and PC5).

The AHC, reported in Figure 4b, confirms the PCA results. Two classes can be clearly distinguished: samples collected at REU and samples collected at PUY, with the exception of 02/10/2019, which is more similar to samples collected at REU. In samples collected at REU, R9 and R10B have a similar composition, slightly different from R8. This could be due, as discussed
in Section 3.3, to the trajectory of the air masses before collection, which are quite similar for R9 and R10B and different for R8, with an intrusion of air from the upper troposphere. They are largely different from 02/10/2019, although they are part of the same cluster. Sample collected at PUY on 02/10/2019 is characterized by an air mass coming from the North and mainly travelling in the boundary layer. Regarding samples collected at PUY, there is a clear separation between winter samples, collected in March, and summer-autumn samples.






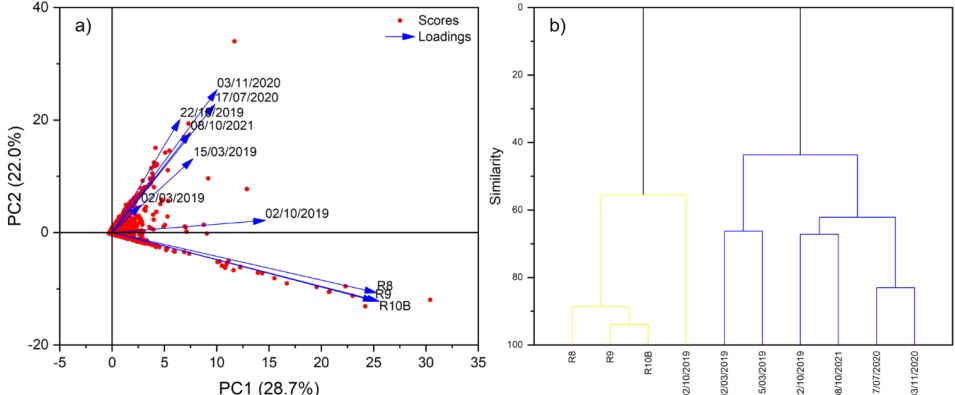

**Figure 4**: a) Bi-plot of the principal component analysis (PCA) performed on samples collected at PUY and at REU. Loadings are the samples, while scores are the molecular formula, represented by red dots. b) Dendrogram obtained for the agglomerative hierarchical clustering (AHC) of the same matrix used in the PCA.

## 510  4 Conclusions

In this study, we present for the first time the molecular characterization of three cloud samples collected in the Southern Hemisphere during the BIO-MAIDO field campaign, in the Réunion Island (REU) (Indian Ocean). Cloud water samples were pre-concentrated by solid phase extraction and analyzed by FT-ICR MS in electrospray negative ionization mode. Mass spectra were treated with Data Analysis and the assignment was performed with MFAssignR software. The composition of samples
collected at the Reunion Island was compared to those of samples collected at puy de Dôme (PUY), in the Massif Central region of France. The same methodology was used for sampling, analysis and data treatment (assignment), in order to obtain a unique database of highly comparable samples. The average elemental composition is similar for the two samples dataset: nevertheless, REU samples present a higher occurrence of CHOS and a lower occurrence of CHNOS compared to PUY samples. This may be due to the marine emission around the island. Other differences, mainly related to cloud processing, are
evidenced by the carbon oxidation state (OSC). This parameter shows similar values for samples collected at REU and at PUY during autumn, but values for winter and summer samples are significantly different. We emitted the hypothesis that, for autumn samples, strong emissions are rapidly processed at REU, due to the high temperature, and aged air masses are collected at PUY, leading to similar values of average OSC, but produced by different causes. The Rivas-Ubach classification shows that 50% of the dissolved organic matter in cloud water accounts for reduced molecules, grouped into the class LipidC. These
compounds are probably linked to emissions from vegetation and urban areas. We calculated that fatty acyls constitute a significant fraction of LipidC (from 11 to 46% in weight). Samples collected at REU have an occurrence of molecular formulas attributed uniquely to fatty acids (12.7%) significantly higher than PUY samples (6.7%). This is probably due to the contribution of sea salt, which is rich of surface active molecules, also called acid-soap complexes. The contribution of the other categories is not significantly different between REU and PUY. 1.3 to 4.0% in weight of the total abundance is
represented by isoprenoids, which contain secondary organic aerosols (SOA) and aqueous SOA derivatives. These formulas were identified in the molecular formula lists of each sample. The statistical tools principal component analysis and agglomerative hierarchical clustering were used to confirm the different composition of samples collected at REU and at PUY. The statistical study evidenced that R8 has a slightly different composition from R9 and R10B, probably due to the trajectory of the air masses before collection, with an intrusion of air from the upper troposphere. For samples collected at PUY, there is
a clear separation between winter and summer-autumn samples, with the exclusion of the sample collected at PUY on



02/10/2019. It is characterized by an air mass coming from the North and mainly travelling in the boundary layer, making it more similar to those collected at REU and charged of fresh emissions. FT-ICR MS is a powerful tool to investigate cloud water composition: in contrast to the methodology previously used for cloud water study, it provides a global detailed overview of the DOM. Combined with classification methodologies, such as the Rivas-Ubach classification (Renard et al., 2022), it may

highlight which families of compounds are preferentially transformed.

**Data availability**

Data are available in the Supplement, but further information can be obtained on request.

**Supplementary information**

The supplement related to this article is available online at:

**Author contributions**

ML and PT are the principal investigators of the BIOMAÏDO project who designed the field campaign and contributed to scientific discussion. LD, MR, J-M P, MV and FB collected the samples at Reunion and at puy de Dôme, LP, HL, IS, EN, MH and AB performed the analysis by FT-ICR MS, PD and AB did the physicochemical characterization of the samples, LP, AB and LD treated and analyzed the data, LP, AB and LD wrote the paper. All the authors reviewed the paper and approved the

final version.

**Competing interests**

The contact author has declared that neither they nor their co-authors have any competing interests.

**Funding**

This work was funded by the French National Research Agency (ANR) thanks to ANR18-CE0-0013-01, also performed in

the framework of the CAP 20-25 Clermont Auvergne Project. The "Observatoire de la Physique du Globe de Clermont Ferrand" and the instrumented site Cezeaux-OPME-puy de Dôme (COPDD) funded the sampling and physicochemical characterization of samples collected at the puy de Dôme observatory. CEA (contract number: CEA CAJ_18-100/C34067 – CNRS 217485) funded the PhD contract of L. Pailler.

**Acknowledgments**

Access to the CNRS research infrastructure Infranalytics (FR2054) is gratefully acknowledged. The authors gratefully acknowledge the team working on BIOMAÏDO field campaign. Thanks to the technical and logistical support provided by "Observatoire des Sciences de l'Univers de la Réunion" and "Laboratoire de l'Atmosphère et des Cyclones". ACTRIS France is also acknowledge for facilitating the access to national facilities, such as the Maïdo observatory.

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
