# Peer review of "Molecular composition of clouds: a comparison between samples collected at tropical (Réunion Island, France) and mid-north (puy de Dôme, France) latitudes."

_EGUsphere, 2023_

## Referee Comment (RC2)

Overview

Molecular composition of clouds: a comparison between samples collected at tropical (Réunion Island, France) and mid-north (puy de Dôme, France) latitudes.

Pailler et al.

**General Comments**

The purpose of this paper is to describe the molecular composition of dissolved organic matter in cloud water at a novel site in Reunion Island and compare it to Puy de Dome in France using primarily FT-ICR MS analysis. The samples are also compared to previous studies an use various metrics to evaluate the composition of the samples for comparison.

Overall, I feel this is a good paper that lays good groundwork for the analysis of cloud water in remote areas that have not previously been investigated with this type of analysis. There are some things that I am interested in and things that should be addressed before full publication, but they are relatively minor and should not hinder its publication in my view.

**Specific Comments**

1.      Line 177: MFAssignR also incorporates H2O, CH2O, and O homologous series for formula extension.

A citation of the package on GitHub, or the manuscript itself (Schum et al. Env. Res. 2020) would be a good addition to this section as well.

2.      Line 274-275: Is there an explanation for why 22/10/2019 has so few MF compared to 8/10/2019? Or maybe why 8/10/2021 has so many more than the rest of the samples? It seems like the DOC is pretty similar between them, with the main differences coming from the inorganic ions. Do you think it is related to the actual sample itself, or to the blank subtraction method? Conservative blank subtraction is a good choice, but I am curious what the formula numbers looked like prior to blank subtraction and whether they were more similar at that point.

3.      Lines 303-304: You mention that the average OSC is similar between PUY and REU autumn samples, while this can definitely just be a coincidence (considering the different sources and conditions) I was curious if you looked into the molecular formulas to see what sort of differences occurred in them. For example is the OSC heavily influenced in both cases by a common set of molecular formulas (even if they are different molecules) or are there really no similarities at all, they just happen to average out to the same OSC?

4.      Lines 362-364: If I am understanding correctly, the general percentage of formulas in each classification is similar between REU and PUY, which seems reasonable, I am still curious about the specific differences between the molecules in one sample or another in a more comprehensive view. Do the formulas in each classification match each other between the different sites or are they largely different? For example, for the LipidC classification, are the formulas found at REU and PUY 90% common, 70%, 50%, less? I think it could be interesting to see if the detailed composition of these samples is very different or the same, since it may say something about the cloud processing results. The "averages" are very useful, but as you have mentioned, even the same formula doesn't necessarily mean

the same molecule, so if a set of molecular formulas are in a particular classification, they may not be similar in any other way, or they could be very similar and highlight that cloud processing brings organic matter to a similar specific result.

5.      Lines 370: While the FT-ICR is very well suited and effective for this work, the lack of structural information is a shortcoming as noted here, is there any interest in doing LC or fragmentation analysis in the future for these samples or others?

6.      Lines 378-384: You are taking appropriate caution in classifying these molecules as one specific class or another with the database, but I was curious whether if you took a few of the formulas that you have classified as "prenol lipids" for example and just looked for any molecule matching that formula (in other databases or the search engine of your choice) if you could get any other classification?

7.      Lines 426-428: I do not quite understand this sentence. Are the measured concentrations for alpha pinene 0.5, 71.5, and 2 for R8, R9, and R10B, while the beta pinene concentrations were 39.9 and 1.3 for R8, R9, and R10B, or are the detection limits for alpha pinene 39.9 and for beta pinene they are 1.3? I think the sentence could be restructured for clarity.

8.      Lines 457: Does this mean that the organosulfate intensity was low in all samples (REU and PUY) with the exception of PUY 8/10/2021, or are you just comparing PUY 08/10/2021 to other PUY samples?

Additionally, you explain the higher occurrence of limonene organosulfates at REU by the increased emission of limonene at the site, which makes sense, but does that imply that the organosulfate formation from limonene is a faster process than the oxidation of pinene? My understanding of the reason given for the relative lack of pinene oxidation products is that the emissions were too fresh to have oxidized yet. Is the organosulfate a primary oxidation product like $C_8H_{12}O_5$? Or is the explanation that there is more limonene emissions relative to the pinenes?

9.      Lines 465: What were the N and S beta caryophalene formulas? Is there any way to know that the formulas are N or S caryophyllene molecules other than matching the formulas? While presence of their emission sources on the coast may explain the N and S beta caryophyllene, why would there be no CHO oxidation products? Are the N and S reactions that much more favorable than the O oxidation? Or is the concentration of N and S so overwhelming that the O oxidation doesn't really occur, relative to N and S?

10.     Line 524: According to the classification you say that 50% of the molecules observed are reduced, is the explanation that the organic matter in the clouds is fairly fresh and hasn't had a chance to oxidize more completely yet?

**Technical Corrections**

11.     Line 25: Somewhat contradictory statements, can consider changing the language a bit to get to the assumed intended meaning.

12.     Line 179: It may be more consistent and precise to say "same mass" instead of "same peak", since the parenthetical on line 180 says "unique mass".

13.     Lines 323: Should probably change "is" to "are"

14.     Lines 520: Instead of "emitted" you should probably say something like "developed" or "produced". Overall the language in this manuscript is very good, but there are few minor things, like this and the comments for lines 426-428 that could be adjusted.

---

## Author Comment (AC1)

**Reviewer 1**

General comments:

The work "Molecular composition of clouds: a comparison between samples collected at tropical Réunion Island, France) and mid-north (puy de Dôme, France) latitudes." by Lucas Pailler et al, represents an important and innovative study about organic compounds in cloud water samples. As the authors pointed out, most studies about cloud water chemical composition in literature are limited to targeted approaches, only covering a relatively small fraction of the organic matter. The authors used ESI-FTICR-MS for a comprehensive non-targeted analysis of DOM in cloud water samples from two different locations.

We would like to acknowledge the reviewer for the accurate analysis of the manuscript and for her/his constructive comments and suggestions. Indeed, the reviewer pointed out some aspects that need to be clarified to improve the quality of the paper.

Many aspects regarding the technical quality of the experimental work and data analysis are excellent. The sample collection and handling process was carefully conducted in respect of the challenging nature and sensibility of environmental samples, which is crucial for a conclusive analysis. Regarding the FTICR-MS measurements, published and proven concepts for data filtering, assignments of elemental composition, graphical representation and interpretation were applied. However, there are several aspects that must be discussed or clarified (see also detailed comments below). I think it is very unfortunate that two different Instruments were used for the analysis of the samples (REU: 12 Tesla FTICR in Rouen; PUY: 9.4 Tesla FTICR at LCM lab). Most of the manuscript outline, scientific question, and discussion is centered on differences between REU and PUY samples. Therefore, the lack of a clear proof (e.g. control samples analyzed on both systems), that measurements between both systems are comparable, is somewhat problematic for such a complex analytical technique. Many experimental parameters can significantly influence obtained mass spectra (e.g. relative abundances of peaks), which could potentially hinder a direct comparison. This uncertainty is amplified by the lack of analytical replicates or an alternative demonstration of method robustness, which would help to understand the significance of the results.

The reviewer pointed out the main problem of this work: samples collected at REU were analyzed on a different FT-ICR MS than samples collected at PUY. This is due to misadventures that are a result of the COVID lock downs and a glitch of the 9.4T-FT-ICR MS. In fact, the group working on clouds (Pailler, Bianco, Deguillaume) usually works with the 9.4T of LCM lab (Bianco et al., 2018, 2019b, a; Pailler et al., 2022). Nevertheless, problems on this instrument made it unavailable until the intervention of a technician and a huge funding to change a part of the instrument. This setback, started in 2020, was additionally slowed down by COVID lock downs. In 2021, the instrument was still unavailable and we needed to analyze the samples of the campaign at REU, which took place in the beginning of 2019. Thus, we decided to use another instrument available in the Infranalytics network, the 12T-FT-ICR MS from COBRA laboratory, using similar settings for the electrospray ionization source. We should highlight that the solid phase extraction (SPE) is the most uncertain step of our analysis and we already show that the discrepancies added by SPE process on three replicates of the same cloud water sample are larger than those added by the replicate of analysis in the same FT-ICR MS (Pailler et al., 2022). Similar formula assignment routines were found applicable in both instruments, and the number of assigned molecular formulas with the 12T-FT-ICR MS from COBRA laboratory were found within the range of our previous results obtained with the

9.4T-FT-ICR MS. When the 9.4T-FT-ICR MS was repaired in 2021, we performed the analysis of the PUY samples.

The overall technical quality of presentation is on a high level and the precise language, style, and literature referencing is very appreciated. A common theme in this work is the classification of the samples based on the sampling location (PUY and REU) to make general conclusions about the difference in their chemical nature. However, the sample set seems not appropriate for this kind of discussion: The three samples for REU were taken within a week, showed comparable number of assigned MFs (2276-3098) and have a very similar organic fingerprint (evidenced by figures 1 b/c, 2 a/b, 3b and figure S6). In contrast, the group of PUY samples were taken over a period of 2.5 years, vary drastically in the number of assigned MFs (120-7436 !) and show very different molecular composition in the OSC vs #C plots (figure 2 c and S4). Therefore, the grouping and comparison of average values calculated over PUY vs REU to differentiate and draw conclusions on the sampling location, does often not seem meaningful (clearly visible in figure 2d, with large boxes for PUY samples due to the heterogeneity of the samples compared to very focused boxes for REU). Instead a more detailed discussion of mass spectral features in individual samples would be interesting, since e.g. the six PUY samples seem to show very different mass spectra.

The reviewer's comment is pertinent and we understand her/his concern. In fact, clouds are atmospheric events quite difficult to sample: to obtain a clean sample, suitable for FT-ICR MS analysis, sampling cannot be automatic and it is very difficult to collect enough volume. Reunion Island has a long term atmospheric observatory at the summit of Mt. Maïdo, but clouds often evaporates at this location. This is the reason why cloud sampling was performed at "piste Omega" for the duration of the campaign. That means that people were available for sampling, on an event basis when clouds were forming on the slope. A longer sampling period, comparable to the one presented for PUY, is not doable at present in REU. Besides, we could argue that, in a tropical location, the emissions from the sea and the vegetation are less variable than on a continental site. The boxplots obtained for clouds collected at Reunion, at least at this sampling location, in the boundary layer and just above the canopy, will always be less variable than those obtained for samples collected at puy de Dôme. Moreover, the composition of the cloud samples collected during the campaign at the Reunion Island and presented in Dominutti et al. (2022) shows low variations of the concentration of main inorganic anions and cations, carboxylic acids, sugars and amino acids.

Sampling at Reunion Island was performed during a field campaign, which is the most common way to study clouds (FEBUKO and HCCT campaigns at Schmucke Mt, …) (van Pinxteren et al., 2005; van Pinxteren et al., 2016; Gioda et al., 2011; Hitzenberger et al., 2002) On the other hand, we sample clouds regularly on long periods at the puy de Dôme station to understand the impact of the seasonal variability and of the air mass origin on the molecular composition. Samples collected at PUY have a very high variability, because clouds are collected in the free troposphere and influenced by long range transport. Two samples collected in the same day shows very different molecular composition, as highlighted by Bianco et al. (2019).

Other articles on FT-ICR MS of cloud samples report two samples (collected on February 24-25th 2010, (Zhao et al., 2013) and 1 month apart, (Bianco et al., 2018)), three samples (collected between November 18th and 21st 2020, (Sun et al., 2021)), six samples (collected in 2017, (Bianco et al., 2019a)) and eight samples (collected in August-September 2014, (Cook et al., 2017)). This is the first dataset of cloud water samples containing ten samples from two different locations.

We are conscious of the limitations of this comparison. Nevertheless, the separate presentation of three cloud samples from Reunion and seven cloud samples from PUY would not be satisfactory for this first characterization of the specificity of the molecular composition of tropical clouds in REU.

To warn the readers on the potential bias of the comparison presented in this work, we added a sentence in the text.

Lines 94-96: "Despite the limitation produced by the analysis of samples collected over different times ranges and with two different FT-ICR MS, we decided to compare the composition of the samples to highlight the differences and similarities of cloud water DOM collected in different environmental conditions but with the same analytical procedure."

Overall, this work has great potential to improve our understanding of DOM in cloud water and many aspects in this manuscript are excellent. However, some details in the experimental setup (FTICR-MS analysis at different instruments, small dataset, no replicates or control samples) hold back the significance of obtained results. This study would furthermore benefit from changing the focus of and intensify the discussion of observations in individual samples instead of mostly a PUY vs REU comparison, since some of the pronounced features in the data are currently ignored.

The reviewer suggests to focus our attention on the observations of individual samples collected at Reunion Island and at puy de Dôme. For puy de Dôme samples, we already presented for two samples the elemental ratios, DBE, aromaticity index and discussed the composition using the van Krevelen classification in Bianco et al. 2018. Then, we analyzed the molecular composition and the presence of secondary organic aerosols tracers using Wozniak classification in Bianco et al. 2019 on six samples, collected at puy de Dôme. Lately, we tested APPI positive ionization and for six samples and the classification presented by Rivas-Ubach (Renard et al., 2022). In our last work, we set-up in detail the methodology of the analysis to make the comparisons as reliable as possible.

Originally, we wanted to present the observations for the three samples collected at Reunion, to give the reader a first glance of the composition of clouds collected at this very peculiar location. Such presentation, however would undermine the novelty of these results as there are several previous studies published on similar topics, already presented by Zhao et al. (2013) for the StormPeak Laboratory, by Cook et al. (2017) for Whiteface mountain and by Sun et al. (2021). To improve the novelty of this work, we decided to compare samples collected at two very different locations.

We think that focusing the attention just on the characterization of three samples collected at the Reunion is not worth for a single paper. For samples collected at puy de Dôme, the presentation of the observations is already published in the previous works of our group.

Specific comments:

P1-L40: The assumption that the difference in cloud water chemical composition between REU and PUY samples is "mainly linked" to chemical processing is not sufficiently supported by the results in my opinion.

We agree with the reviewer, primary emissions have a crucial role. The sentence was modified as follow:

Lines 39-41: "Samples collected at REU, have a different composition from samples collected at PUY, mainly linked to different primary sources and to the processing of organic matter in cloud water, but also to the influence of different primary emissions at the two locations."

P4-L145: While the explanation for the number of assignments (higher R => more peaks, ion suppression => less peaks, + different chemical composition) might be theoretically correct it is not convincing for the presented data. The actual data does not really follow any trend. Specifically: 12T instrument: quite consistent for the three samples ranging between 2000-3000 MFs. 9.4T instrument: 120 (!) – 7000 (!) assigned peaks. Also, a comparison to DOC values (vs. ESI-FTICR-MS response) would be appreciated, as DOC is relatively similar for all samples, while the MS response seem to vary in more than one magnitude of order. A better explanation why the ion suppression effects are influencing specifically the three mentioned samples would also be interesting. And: Is it actually reasonable to include them in the analysis/comparison if their MS quality might not be sufficient?

The composition of atmospheric samples is very variable and largely depends on the air mass origin of the samples. As explained above, the long term cloud collection at the puy de Dôme station revealed a very high variability in the composition of cloud aqueous phase, already evidenced in Bianco et al. (2019). On the other hand, in Reunion Island, the variability of the composition is less pronounced, since the emissions from the Ocean are persistent and the emissions from vegetation, which is always verdant with low seasonal variability, are presumably comparable during the year. Less seasonal variations are expected also for the anthropogenic activity. To summarize, we could argue that the molecular composition of cloud samples collected in Reunion Island (in the boundary layer) is less variable than the variability observed at puy de Dôme. This would lead to a very different number of molecular formula for samples collected at puy de Dôme and to less variations for samples collected at Reunion Island. We think that the variability is an important parameter that needs to be included in the comparison, to have a dataset representative of the environmental conditions.

The following figure (Fig. 1) reports the number of MF obtained with ESI FT-ICR MS analysis vs the DOC concentration for samples presented in this work. Samples presented in Bianco et al 2018 and 2019 (9.4 T, Bruker Solarix), Cook et al., 2018 (12T Bruker SolariX), Zhao et al., 2013 (LTQFTUltra, ThermoScientific), and Sun et al. 2021 (9.4-T Bruker SolariX XR) are also presented in the plot. It's worth noting that no trend is observed between the DOC concentration and the number of MF in the ESI FT-ICR MS response. As noted before, samples from the PUY show a huge variability in the number of MFs. Moreover, the work of Cook et al. 2017 also shows a huge variability in the number of MFs for samples in the same range of DOC concentration (between 430 and 2300 MF). The variability and lack of trend existing in the published cloud samples shows the quality of the analysis cannot be evaluated just comparing the DOC with the number of MFs.

This interesting result is now reported in the text.

Lines 285-288:" "This variability is due to the influence of primary emissions and to the atmospheric reactivity: a similar variability in the number of MFs has been already observed in previous works (Zhao et al., 2013; Cook et al., 2017; Sun et al., 2021; Bianco et. al., 2018; Bianco et al., 2019). In addition, the DOC concentration and the number of MFs show no correlation in this work as well as in published data (Figure S1)."

We agree with the reviewer that the ions suppression effects, alone, cannot explain the variability of the samples, which is mostly linked to the environmental variability. The sentence was modified in the text.

Lines 187-194: "The instrument with higher resolving power was expected to enable the assignment of a larger number of assigned molecular formulas within the chosen margin of tolerance, as more isobaric

species were separated. Using a similar tuning for 12 T and 9.4 T FT-ICR MS, and similar formula assignment routines, the number of assigned molecular formulas were found to be different depending on the sample and not only on the resolving power of the instrument. In fact, depending on the chemical composition of the sample and the SPE step, ion suppression effects could occur with ESI, thus decreasing the overall number of interpretable signals. To atone for this difference in total numbers of assigned molecular formulas, we chose to use a conservative approach by working with relative number of occurrence (or relative weighted occurrence) of molecular formulas within a class (equations 1 and 2 in S1 in the Supplementary information, hereafter SI)."

[Figure]

Figure 1: Correlation between the number of MFs and the dissolved organic carbon concentration in cloud samples.

P4-L149: The terms "relative number of occurrence" or "relative weighted occurrence" is not used in the cited publication Hawkes et al 2020. I can guess that the data was normalized and/or weighted to the total ion current to give the relative abundance instead of using absolute ion counts?

We agree with the reviewer, these terms should be more precisely defined. The relative number of occurrence is calculated as the number of MFs in a specific group divided by the total number of assigned MFs and multiplied by 100, as reported in equation 1 for CHO compounds.

$$\text{Equation 1: } relative\ number_{CHO} = \frac{\#MFs_{CHO}}{\#MFs_{Total}} \times 100$$

The relative weighted occurrence was calculated as the sum of the absolute intensities of a specific group, divided by the sum of the absolute intensities of all the assigned MFs and multiplied by 100, as reported in equation 2 for CHO compounds:

$$\text{Equation 2: } relative\ weighted_{CHO} = \frac{\sum absolute\ intensity_{CHO}}{\sum absolute\ intensity_{Total}} \times 100$$

We decided to avoid the use of total ion current (TIC) because less than 60% of the TIC is assigned (due to restrictive rules and blank exclusion), and the normalization by the TIC would complexify the comparison and the discussion.

The sentence was modified as follow:

Paragraph S.1: "The relative number of occurrence is calculated as the number of MFs in a specific group divided by the total number of assigned MFs and multiplied by 100, as reported in equation 1 for CHO compounds.

$$\text{Equation 1: } relative\ number_{CHO} = \frac{\#MFs_{CHO}}{\#MFs_{Total}} \times 100$$

The relative weighted occurrence was calculated as the sum of the absolute intensities of a specific group, divided by the sum of the absolute intensities of all the assigned MFs and multiplied by 100, as reported in equation 2 for CHO compounds:

$$\text{Equation 2: } relative\ weighted_{CHO} = \frac{\sum absolute\ intensity_{CHO}}{\sum absolute\ intensity_{Total}} \times 100"$$

In any case, there is some confusion regarding the normalization & discussion of the data since the authors also use absolute counts for argumentation several times within this manuscript (which actually might not be viable for this specific analysis).

Relative abundance based on absolute counts is currently used by other groups: Zhao et al. (2013) calculated weighted O/C, H/C and DBE using the relative abundance based on the number of counts; the same calculation was applied by Bianco et al. (2018). Sun et al. (2023) calculated the weighted abundance characteristic parameter (X) as $X=\Sigma(X_i \times Int_i)/\Sigma Int_i$, where $X_i$ is the parameter of interest (O/C, DBE, ..) and $Int_i$ is the intensity of the formula i.

P4-L191: I am wondering how "not-found" MFs were treated. I guess the matrix of 9251 MFs represents all MFs, which were found in the whole dataset and for samples a given MF was not detected, a value of "0" was defined? I could imagine the statistical analysis can be very distorted for samples which have a low number of found MFs. E.g. for sample 22/10/2019 there are 120 variables with an actual numerical value between 0-1, potentially even dominated by very few signals, while more than 9000 (equally treated) scores are "0". Also, especially for the statistical analysis I am suspecting a large influence of using different MS-instruments.

As the reviewer guessed, a value of 0 was defined for non detected MFs. Gurganus et al. (2015) report in their manuscript

"A total of 7029 unique molecular formulas, representing all of the formulas assigned from the peaks in the FTICR-MS datasets, were input as the variables for the PCA. The relative spectral intensity of each of the peaks within a given mass spectrum is measured as the spectral intensity of that peak divided by the summed relative intensity of every peak assigned a molecular formula in that mass spectrum. These relative intensities were determined for each of the spectra for the 17 size fractions and were used to generate the PCA matrix, with a score of zero given for molecular formulas not present in a fraction and the relative intensities within each fraction summing to one."

We adopted the same methodology to set-up the matrix for the PCA and AHC.

If we decide to discard sample 22/10/2019, we should define a threshold to decide which samples we should treat or not. We agree with the reviewer that samples with a low number of MFs are not well represented in the statistical analysis but we think that excluding them would lead to a total non-representation of samples with a low variety in organic matter.

Concerning the influence of using different MS instruments, it should be indeed mentioned. A sentence was added to the text.

Lines 204-205: "Unluckily, samples were analyzed with two different FT-ICR MS, which could potentially introduce a bias in the statistical analysis."

P9-L295: The use of ANOVA in this dataset for a strict PUY vs REU differentiation seems forced and not meaningful. According to figure 2a: two PUY samples (02/03/2019 and 15/03/2019) have a lower average OSC than REU samples, two PUY samples (02/10/2019, 17/07/2020) have a higher OSC and the rest of the PUY samples have comparable OSC values (as also mentioned later by the authors). Have you tried to check the significance of other groupings (seasons, air mass history)?

We used ANOVA in its simplest form to perform a statistical test of whether two or more population means are equal and thus to understand if the differences observed were statistically significant or not within all the samples. Seasonal variations are discussed starting at line 300 on page 9. The t-tests were performed to check the differences between Reunion and samples collected at PUY in different season and are summarized in the text and in the following table.

Table S3. Average, standard deviation and t-tests of OSC collected at Reunion, and winter, summer and autumn at PUY.

|  | mean | std dev | t test with Reunion | Significantly different? |
|---|---|---|---|---|
| mean winter | -0.97 | 0.56 | 3.36E-93 | Yes |
| mean summer | -0.60 | 0.58 | 7.99E-03 | Yes |
| mean autumn | -0.75 | 0.65 | 0.64 | No |
| mean Reunion | -0.77 | 0.55 |  |  |

Two sentences were added in the text:

Lines 313-314: "The t-test was used to confirm these results using the average and standard deviation of OSC."

Lines 316-317: "The t-test showed that PUY samples collected in winter and summer are statistically different from those collected in REU."

Actually, the whole significance calculation for the OSC might not be meaningful in the way it seems to be conducted: the uncertainty of the OSC appear to be calculated by taking the variation over all individual MFs within a sample (similar to how the boxplot 2a presents the data). That's why they are so high (e.g. -0.97±0.56 for winter). Basically, this value describes the range of how different the OSC of individual analytes in the sample is, instead of how different the average OSC value of samples within a group is.

Maybe some clarification on the significance calculation would be helpful to better understand the approach.

We are sorry, but we do not understand the comment of the reviewer and how we can improve our methodology. OSC values are calculated using the formula $OSC = -\Sigma_i$ (oxidation state$_i$ x $n_i$ / $n_C$), as reported by (Kroll et al., 2011), and currently used in atmospheric studies. Figure 2 reports the boxplot where the box represents the $1^{st}$-$3^{rd}$ quartile, the continuous line is the median, the dot is the mean and the whiskers the $10^{th}$ and $90^{th}$ percentiles.

P9-L322: The interpretation of Figure 2b/c is missing some important observations. E.g. the fact that PUY samples are often dominated by few very abundant signals (e.g. for 02/10/2019: very large bubbles at 15-20 carbon atoms and 5-10 carbon atoms). Also visible for figure S4 with huge bubbles for 02/03/2019 which are not discussed and are significantly different from the other "summer" sample 15/03/2019.

We agree with the reviewer. A sentence was added to the text.

Lines 333-338: "REU samples present a very homogeneous signal, with OSC between -2 and 1 and nC higher than 5. Autumn samples from PUY are often dominated by few signals with high intensity, as in the case of sample 02/10/2019, which shows an intense signal at nC range of 5-10. Less signal is present for nC higher than 25, except for 08/10/2021. Winter and summer samples collected at PUY show different characteristics, with a high number of signal with high intensity at nC higher than 20 for winter samples and a more intense signal for nC lower than 10 for the summer sample."

Also, the scaling in figure 2 (and S4) is hindering a proper interpretation. Bubble sizes are proportional to the absolute intensity (while in other instances in the manuscript relative intensities are used). Its not possible to get an idea of the chemical space for samples 03/11/2020 and 22/10/2019. This is relevant to better compare the other figures and statistical analysis (which are normalized to relative intensities).

[Figure]

Bubbles for 2b should also be transparent. In its current form its not possible to see and evaluate the chemical composition of R8 and R9. But it seems MFs are relatively homogenously distributed over the chemical space for all three samples, in contrast to PUY samples.

Figure 2 and figure S4 have been modified according to reviewer's comments.

P9-L325: I agree that one of the most significant differences between PUY and REU is the lower right region from OSC -1.5 to -4 and 0 to 10 carbon atoms. However, I don't understand the conclusion "compounds in this region can be recalcitrant to oxidation in cloud water" (means hard to oxidize? Is there a reference for this statement?) and "These results clearly assess that DOM in REU samples is less oxidized than DOM in PUY samples". Are these two statements directly connected? Or is the second statement a summary of the whole paragraph? In any case, a more detailed interpretation of the region in the lower right would be important, since it seems very significant for PUY samples.

According to the reviewer's comment, the sentence "compounds in this region can be recalcitrant to oxidation in cloud water" has been deleted, since we have no sufficient data to draw this conclusion. The following sentence is not connected with the previous one and it's simply a summary of the whole paragraph. This sentence was also deleted from the text.

We agree with the reviewer on the potential importance of the lower right region of the plot for the PUY samples. The compounds in these region were investigated more in deep. 120 different formula were found, mainly composed by CHNOS (80%) and CHNO (8.4%). All the formula have at least 1 N atom. The average O/C and H/C ratios are 0.65±0.35 and 1.89±0.38, respectively. One formula occurred in all the samples, C5H7N3O3S, while two occurred in 6 (out of 7) samples, $C_8H_{17}N_3O_4S$ and $C_8H_{17}N_3O_5S$. It's difficult to find an interpretation of this region of the OSC-C plot, but these results are now added in the text.

Lines 350-354: " A more detailed analysis of this region of the plot shows that 120 different formula were found, mainly composed by CHNOS (80%) and CHNO (8.4%). All the formula have at least one nitrogen atom. The average O/C and H/C ratios are 0.65 ± 0.35 and 1.89 ± 0.38, respectively. One formula occurred in all the samples, $C_5H_7N_3O_3S$, while two occurred in 6 (out of 7) samples, $C_8H_{17}N_3O_4S$ and $C_8H_{17}N_3O_5S$. "

P12-L415: The Interpretation for the lack of organosulfates (L415-424) is very hard to follow and appears to be mostly speculation. Most of the paragraph is not connected to observations from this study. For the reader it's hard to follow why literature values for isoprene (Dominutti et al., 2022 and Wang et al., 2020) are discussed and how they relate to the samples from this study (especially since table 1 lists isoprene as a target compound for this present study).

We understand the reviewer comment: the sentence was removed from the text.

P14 -L479: Regarding typical MFs for PUY and REU: PC1 and PC2 seem to clearly differentiate between PUY and REU samples. There are also several MFs on the scores plot that align well with the R8, R9, and R10B loading vectors. What are the MFs mostly related to PC1 and PC2? It would be interesting to identify & extract these MFs and present them e.g. in an OSC vs #C space since they seem to be the relevant observations for the discrimination? This could also give a chemical meaning to the statistical analysis. At the moment it mainly serves to show that mass spectral fingerprints (also keeping in mind that these were measured on different instruments) are different.

The reviewer is right, a more in deep analysis of the results of the PCA would be great to understand the variability of the composition with the sampling site and season. However, to perform this analysis in the correct way, we should inverse the matrix (samples should be the objects and not the variables). Moreover, more samples should be added to the matrix to make the analysis more reliable. As the reviewer pointed out in the previous paragraphs, we should also improve the treatment of absent molecular formulas.

We plan to continue in this direction, but the results of this statistical analysis will be part of a new project (OPTIC, funded by ANR research program) and will be presented in a further publication.

**References**

Bianco, A., Deguillaume, L., Vaïtilingom, M., Nicol, E., Baray, J.-L., Chaumerliac, N., and Bridoux, M. C.: Molecular Characterization of Cloud Water Samples Collected at the puy de Dôme (France) by Fourier Transform Ion Cyclotron Resonance Mass Spectrometry, Environmental Science & Technology, https://doi.org/10.1021/acs.est.8b01964, 2018.

Bianco, A., Riva, M., Baray, J.-L., Ribeiro, M., Chaumerliac, N., George, C., Bridoux, M., and Deguillaume, L.: Chemical Characterization of Cloudwater Collected at Puy de Dôme by FT-ICR MS Reveals the Presence of SOA Components, ACS Earth Space Chem., 3, 2076–2087, https://doi.org/10.1021/acsearthspacechem.9b00153, 2019a.

Bianco, A., Deguillaume, L., Chaumerliac, N., Vaïtilingom, M., Wang, M., Delort, A.-M., and Bridoux, M. C.: Effect of endogenous microbiota on the molecular composition of cloud water: a study by Fourier-transform ion cyclotron resonance mass spectrometry (FT-ICR MS), Scientific Reports, 9, https://doi.org/10.1038/s41598-019-44149-8, 2019b.

Cook, R. D., Lin, Y.-H., Peng, Z., Boone, E., Chu, R. K., Dukett, J. E., Gunsch, M. J., Zhang, W., Tolic, N., and Laskin, A.: Biogenic, urban, and wildfire influences on the molecular composition of dissolved organic compounds in cloud water, Atmospheric Chemistry and Physics, 17, 15167–15180, 2017.

Dominutti, P. A., Renard, P., Vaïtilingom, M., Bianco, A., Baray, J.-L., Borbon, A., Bourianne, T., Burnet, F., Colomb, A., Delort, A.-M., Duflot, V., Houdier, S., Jaffrezo, J.-L., Joly, M., Leremboure, M., Metzger, J.-M., Pichon, J.-M., Ribeiro, M., Rocco, M., Tulet, P., Vella, A., Leriche, M., and Deguillaume, L.: Insights into tropical cloud chemistry in Réunion (Indian Ocean): results from the BIO-MAÏDO campaign, Atmos. Chem. Phys., 22, 505–533, https://doi.org/10.5194/acp-22-505-2022, 2022.

Gioda, A., Reyes-Rodríguez, G. J., Santos-Figueroa, G., Collett, J. L., Decesari, S., Ramos, M. D. C. K. V., Bezerra Netto, H. J. C., De Aquino Neto, F. R., and Mayol-Bracero, O. L.: Speciation of water-soluble inorganic, organic, and total nitrogen in a background marine environment: Cloud water, rainwater, and aerosol particles, J. Geophys. Res., 116, D05203, https://doi.org/10.1029/2010JD015010, 2011.

Gurganus, S. C., Wozniak, A. S., and Hatcher, P. G.: Molecular characteristics of the water soluble organic matter in size-fractionated aerosols collected over the North Atlantic Ocean, Marine Chemistry, 170, 37–48, https://doi.org/10.1016/j.marchem.2015.01.007, 2015.

Hitzenberger, R., Berner, A., Kasper-Giebl, A., Löflund, M., and Puxbaum, H.: Surface tension of Rax cloud water and its relation to the concentration of organic material, J.-Geophys.-Res., 107, https://doi.org/10.1029/2002JD002506, 2002.

Kroll, J. H., Donahue, N. M., Jimenez, J. L., Kessler, S. H., Canagaratna, M. R., Wilson, K. R., Altieri, K. E., Mazzoleni, L. R., Wozniak, A. S., Bluhm, H., Mysak, E. R., Smith, J. D., Kolb, C. E., and Worsnop, D. R.: Carbon oxidation state as a metric for describing the chemistry of atmospheric organic aerosol, Nature Chemistry, 3, 133–139, https://doi.org/10.1038/nchem.948, 2011.

Pailler, L., Renard, P., Nicol, E., Deguillaume, L., and Bianco, A.: How Well Do We Handle the Sample Preparation, FT-ICR Mass Spectrometry Analysis, and Data Treatment of Atmospheric Waters?, Molecules, 27, 7796, https://doi.org/10.3390/molecules27227796, 2022.

van Pinxteren, D., Plewka, A., Hofmann, D., Müller, K., Kramberger, H., Svrcina, B., Bächmann, K., Jaeschke, W., Mertes, S., Collett, J. L., and Herrmann, H.: Schmücke hill cap cloud and valley stations aerosol characterisation during FEBUKO (II): Organic compounds, Atmospheric Environment, 39, 4305–4320, https://doi.org/10.1016/j.atmosenv.2005.02.014, 2005.

Renard, P., Bianco, A., Jänis, J., Kekäläinen, T., Bridoux, M., and Deguillaume, L.: Puy de Dôme Station (France): A stoichiometric approach to compound classification in clouds, JGR Atmospheres, https://doi.org/10.1029/2022JD036635, 2022.

Sun, W., Fu, Y., Zhang, G., Yang, Y., Jiang, F., Lian, X., Jiang, B., Liao, Y., Bi, X., Chen, D., Chen, J., Wang, X., Ou, J., Peng, P., and Sheng, G.: Measurement report: Molecular characteristics of cloud water in southern China and insights into aqueous-phase processes from Fourier transform ion cyclotron resonance mass spectrometry, Atmos. Chem. Phys., 21, 16631–16644, https://doi.org/10.5194/acp-21-16631-2021, 2021.

van Pinxteren, D., Fomba, K. W., Mertes, S., Müller, K., Spindler, G., Schneider, J., Lee, T., Collett, J. L., and Herrmann, H.: Cloud water composition during HCCT-2010: Scavenging efficiencies, solute concentrations, and droplet size dependence of inorganic ions and dissolved organic carbon, Atmospheric Chemistry and Physics, 16, 3185–3205, https://doi.org/10.5194/acp-16-3185-2016, 2016.

Zhao, Y., Hallar, A. G., and Mazzoleni, L. R.: Atmospheric organic matter in clouds: exact masses and molecular formula identification using ultrahigh-resolution FT-ICR mass spectrometry, Atmospheric Chemistry and Physics, 13, 12343–12362, 2013.

---

## Author Comment (AC2)

**Reviewer 2**

General Comments

The purpose of this paper is to describe the molecular composition of dissolved organic matter in cloud water at a novel site in Reunion Island and compare it to Puy de Dome in France using primarily FT-ICR MS analysis. The samples are also compared to previous studies and use various metrics to evaluate the composition of the samples for comparison.

Overall, I feel this is a good paper that lays good groundwork for the analysis of cloud water in remote areas that have not previously been investigated with this type of analysis. There are some things that I am interested in and things that should be addressed before full publication, but they are relatively minor and should not hinder its publication in my view.

We would like to acknowledge the reviewer for the accurate for her/his constructive comments and suggestions. The concerns raised helped us to look more in deep in the data and to improve the quality and the presentation of the results.

**Specific Comment**s

Line 177: MFAssignR also incorporates H2O, CH2O, and O homologous series for formula extension. A citation of the package on GitHub, or the manuscript itself (Schum et al. Env. Res. 2020) would be a good addition to this section as well.

We agree with the reviewer. The reference was added in this section at line 168.

Line 274-275: Is there an explanation for why 22/10/2019 has so few MF compared to 8/10/2019? Or maybe why 8/10/2021 has so many more than the rest of the samples? It seems like the DOC is pretty similar between them, with the main differences coming from the inorganic ions. Do you think it is related to the actual sample itself, or to the blank subtraction method? Conservative blank subtraction is a good choice, but I am curious what the formula numbers looked like prior to blank subtraction and whether they were more similar at that point.

We are persuaded that the very variable number of MFs is related to the sample itself, as shown in the following table that reports the number of MFs before and after blank subtraction for each sample presented in this work. We already observed very different numbers of MFs in cloud samples in Bianco et al., (2019) and in Cook et al., (2017). The following plot represents the DOC vs ESI FT-ICR MS number of MF for samples from the PUY and the REU. Samples presented in Bianco et al., (2018) and 2019 (9.4 T, Bruker Solarix), Cook et al., 2017 (12T Bruker SolariX), Zhao et al., (2013) (LTQFTUltra, ThermoScientific), and Sun et al., (2021) (9.4-T Bruker SolariX XR) are presented in the plot. It's worth noting that no trend is observed between the DOC concentration and the number of MFs in the ESI FT-ICR MS response. As noted before, samples from the PUY show a huge variability in the number of MFs. Moreover, the work of Cook et al. 2017 also shows a huge variability in the number of MF for samples in the same range of DOC concentration (between 430 and 2300 MF).

| Sample | #MF before blank subtraction | #MF after blank subtraction | #MF in common |
|--------|------------------------------|-----------------------------|---------------|
| R8 | 3199 | 3098 | 101 |
| R9 | 2739 | 2503 | 236 |
| R10B | 2463 | 2276 | 187 |
| 02/03/2019 | 3441 | 3244 | 197 |
| 15/03/2019 | 2345 | 2084 | 261 |
| 02/10/2019 | 1555 | 1543 | 12 |
| 22/10/2019 | 222 | 120 | 102 |
| 17/07/2020 | 849 | 715 | 134 |
| 03/11/2020 | 433 | 312 | 121 |
| 08/10/2021 | 8073 | 7436 | 637 |

[Figure]

This interesting result is now reported in the text.

Lines 285-288: "This variability is due to the influence of primary emissions and to the atmospheric reactivity: a similar variability in the number of MFs has been already observed in previous works (Zhao et al., 2013; Cook et al., 2017; Sun et al., 2021; Bianco et. al., 2018; Bianco et al., 2019). In addition, the DOC concentration and the number of MFs show no correlation in this work as well as in published data (Figure S1)."

Lines 303-304: You mention that the average OSC is similar between PUY and REU autumn samples, while this can definitely just be a coincidence (considering the different sources and conditions) I was curious if you looked into the molecular formulas to see what sort of differences occurred in them. For example is the OSC heavily influenced in both cases by a common set of molecular formulas (even if they are different molecules) or are there really no similarities at all, they just happen to average out to the same OSC?

We agree with the reviewer that this point is important and needs more discussion. Nevertheless, it is not easy to find an approach to answer this question.

We started by comparing the molecular formulas in the presented samples through a Venn diagram, which is difficult to represent for the whole dataset. The following table present the different intersections for the Venn diagram and the number of MF for each intersection. Surprisingly, only 8 MFs are common to all

the samples (line 1, in yellow). These formula are CHO compounds, namely $C_{15}H_{26}O_8$, $C_{14}H_{22}O_9$, $C_{11}H_{12}O_4$, $C_{17}H_{26}O_9$, $C_{10}H_{14}O_6$, $C_9H_{16}O_6$, $C_{15}H_{24}O_9$, and $C_7H_8O_5$, with OSC values between -0.18 and +0.77. The number of MFs contained in 9 samples is 46, in 8 samples is 108, in 7 samples is 250, in 6 samples is 586. MFs contained only in one sample are reported in the last ten lines of the table and represent from 5 to 50% of the total number of MFs in the sample.

| Intersection between | | | | | | | | | | #MF |
|---|---|---|---|---|---|---|---|---|---|---|
| *02/03/2019* | *02/10/2019* | *03/11/2020* | *08/10/2021* | *15/03/2019* | *17/07/2020* | *22/10/2019* | *R10B* | *R8* | *R9* | *8* |
| 02/03/2019 | 02/10/2019 | 08/10/2021 | 15/03/2019 | 17/07/2020 | 22/10/2019 | R10B | R8 | R9 | | 9 |
| 02/03/2019 | 02/10/2019 | 03/11/2020 | 08/10/2021 | 15/03/2019 | 17/07/2020 | R10B | R8 | R9 | | 35 |
| 02/03/2019 | 02/10/2019 | 03/11/2020 | 08/10/2021 | 15/03/2019 | 17/07/2020 | 22/10/2019 | R8 | R9 | | 2 |
| 02/03/2019 | 02/10/2019 | 08/10/2021 | 15/03/2019 | 22/10/2019 | R10B | R8 | R9 | | | 6 |
| 02/03/2019 | 02/10/2019 | 08/10/2021 | 15/03/2019 | 17/07/2020 | R10B | R8 | R9 | | | 77 |
| 02/03/2019 | 02/10/2019 | 03/11/2020 | 08/10/2021 | 15/03/2019 | R10B | R8 | R9 | | | 18 |
| 02/03/2019 | 02/10/2019 | 08/10/2021 | 17/07/2020 | 22/10/2019 | R10B | R8 | R9 | | | 1 |
| 02/03/2019 | 02/10/2019 | 03/11/2020 | 08/10/2021 | 17/07/2020 | R10B | R8 | R9 | | | 1 |
| 02/10/2019 | 03/11/2020 | 08/10/2021 | 15/03/2019 | 17/07/2020 | R10B | R8 | R9 | | | 3 |
| 02/10/2019 | 03/11/2020 | 08/10/2021 | 15/03/2019 | 17/07/2020 | 22/10/2019 | R8 | R9 | | | 1 |
| 02/03/2019 | 02/10/2019 | 03/11/2020 | 08/10/2021 | 15/03/2019 | 17/07/2020 | R10B | R9 | | | 1 |
| 02/03/2019 | 02/10/2019 | 08/10/2021 | 15/03/2019 | R10B | R8 | R9 | | | | 162 |
| 02/03/2019 | 08/10/2021 | 15/03/2019 | 22/10/2019 | R10B | R8 | R9 | | | | 3 |
| 02/03/2019 | 03/11/2020 | 15/03/2019 | 17/07/2020 | R10B | R8 | R9 | | | | 1 |
| 02/03/2019 | 08/10/2021 | 15/03/2019 | 17/07/2020 | R10B | R8 | R9 | | | | 6 |
| 02/03/2019 | 03/11/2020 | 08/10/2021 | 15/03/2019 | R10B | R8 | R9 | | | | 7 |
| 02/03/2019 | 02/10/2019 | 08/10/2021 | 17/07/2020 | R10B | R8 | R9 | | | | 7 |
| 02/03/2019 | 02/10/2019 | 03/11/2020 | 08/10/2021 | R10B | R8 | R9 | | | | 1 |
| 02/10/2019 | 08/10/2021 | 15/03/2019 | 17/07/2020 | R10B | R8 | R9 | | | | 6 |
| 02/10/2019 | 03/11/2020 | 08/10/2021 | 17/07/2020 | R10B | R8 | R9 | | | | 6 |
| 02/03/2019 | 02/10/2019 | 08/10/2021 | 15/03/2019 | 17/07/2020 | R8 | R9 | | | | 2 |
| 02/03/2019 | 02/10/2019 | 08/10/2021 | 15/03/2019 | 17/07/2020 | R10B | R8 | | | | 1 |
| 02/03/2019 | 02/10/2019 | 03/11/2020 | 08/10/2021 | 15/03/2019 | R10B | R8 | | | | 1 |
| 02/03/2019 | 02/10/2019 | 03/11/2020 | 08/10/2021 | 17/07/2020 | R10B | R8 | | | | 1 |
| 02/03/2019 | 02/10/2019 | 03/11/2020 | 08/10/2021 | 15/03/2019 | 17/07/2020 | R8 | | | | 2 |
| 02/03/2019 | 02/10/2019 | 08/10/2021 | 15/03/2019 | 22/10/2019 | R10B | R9 | | | | 1 |
| 02/03/2019 | 02/10/2019 | 08/10/2021 | 15/03/2019 | 17/07/2020 | R10B | R9 | | | | 2 |
| 02/03/2019 | 02/10/2019 | 03/11/2020 | 08/10/2021 | 15/03/2019 | 17/07/2020 | R9 | | | | 1 |
| 02/03/2019 | 02/10/2019 | 03/11/2020 | 08/10/2021 | 15/03/2019 | 17/07/2020 | 22/10/2019 | | | | 40 |
| 02/03/2019 | 02/10/2019 | 15/03/2019 | R10B | R8 | R9 | | | | | 5 |
| 02/03/2019 | 08/10/2021 | 15/03/2019 | R10B | R8 | R9 | | | | | 416 |
| 02/03/2019 | 02/10/2019 | 08/10/2021 | R10B | R8 | R9 | | | | | 23 |

| Intersection between | | | | | | | | | | #MF |
|---|---|---|---|---|---|---|---|---|---|---|
| 02/10/2019 | 08/10/2021 | 15/03/2019 | R10B | R8 | R9 | | | | | 7 |
| 08/10/2021 | 15/03/2019 | 17/07/2020 | R10B | R8 | R9 | | | | | 2 |
| 02/10/2019 | 08/10/2021 | 22/10/2019 | R10B | R8 | R9 | | | | | 1 |
| 02/10/2019 | 03/11/2020 | 17/07/2020 | R10B | R8 | R9 | | | | | 3 |
| 02/10/2019 | 08/10/2021 | 17/07/2020 | R10B | R8 | R9 | | | | | 41 |
| 02/10/2019 | 03/11/2020 | 08/10/2021 | R10B | R8 | R9 | | | | | 2 |
| 02/03/2019 | 02/10/2019 | 08/10/2021 | 15/03/2019 | R8 | R9 | | | | | 7 |
| 02/03/2019 | 02/10/2019 | 08/10/2021 | 17/07/2020 | R8 | R9 | | | | | 1 |
| 02/10/2019 | 03/11/2020 | 08/10/2021 | 17/07/2020 | R8 | R9 | | | | | 2 |
| 02/03/2019 | 02/10/2019 | 08/10/2021 | 15/03/2019 | R10B | R8 | | | | | 7 |
| 02/03/2019 | 02/10/2019 | 08/10/2021 | 17/07/2020 | R10B | R8 | | | | | 1 |
| 02/10/2019 | 03/11/2020 | 08/10/2021 | 17/07/2020 | R10B | R8 | | | | | 3 |
| 02/03/2019 | 02/10/2019 | 08/10/2021 | 15/03/2019 | 22/10/2019 | R8 | | | | | 1 |
| 02/03/2019 | 02/10/2019 | 08/10/2021 | 15/03/2019 | 17/07/2020 | R8 | | | | | 1 |
| 02/03/2019 | 02/10/2019 | 03/11/2020 | 08/10/2021 | 17/07/2020 | R8 | | | | | 1 |
| 02/03/2019 | 02/10/2019 | 08/10/2021 | 15/03/2019 | R10B | R9 | | | | | 13 |
| 02/03/2019 | 02/10/2019 | 08/10/2021 | 15/03/2019 | 17/07/2020 | R10B | | | | | 1 |
| 02/10/2019 | 03/11/2020 | 08/10/2021 | 15/03/2019 | 17/07/2020 | R10B | | | | | 1 |
| 02/03/2019 | 02/10/2019 | 08/10/2021 | 15/03/2019 | 17/07/2020 | 22/10/2019 | | | | | 6 |
| 02/03/2019 | 02/10/2019 | 03/11/2020 | 08/10/2021 | 15/03/2019 | 22/10/2019 | | | | | 4 |
| 02/03/2019 | 02/10/2019 | 03/11/2020 | 08/10/2021 | 15/03/2019 | 17/07/2020 | | | | | 36 |
| 02/10/2019 | 03/11/2020 | 08/10/2021 | 15/03/2019 | 17/07/2020 | 22/10/2019 | | | | | 1 |
| 02/03/2019 | 15/03/2019 | R10B | R8 | R9 | | | | | | 71 |
| 02/03/2019 | 02/10/2019 | R10B | R8 | R9 | | | | | | 1 |
| 02/03/2019 | 08/10/2021 | R10B | R8 | R9 | | | | | | 124 |
| 08/10/2021 | 15/03/2019 | R10B | R8 | R9 | | | | | | 21 |
| 02/10/2019 | 17/07/2020 | R10B | R8 | R9 | | | | | | 1 |
| 02/10/2019 | 08/10/2021 | R10B | R8 | R9 | | | | | | 39 |
| 08/10/2021 | 17/07/2020 | R10B | R8 | R9 | | | | | | 8 |
| 02/03/2019 | 08/10/2021 | 15/03/2019 | R8 | R9 | | | | | | 75 |
| 02/03/2019 | 02/10/2019 | 08/10/2021 | R8 | R9 | | | | | | 9 |
| 02/10/2019 | 08/10/2021 | 15/03/2019 | R8 | R9 | | | | | | 4 |
| 02/10/2019 | 03/11/2020 | 17/07/2020 | R8 | R9 | | | | | | 1 |
| 02/10/2019 | 08/10/2021 | 17/07/2020 | R8 | R9 | | | | | | 6 |
| 02/03/2019 | 15/03/2019 | 17/07/2020 | R10B | R8 | | | | | | 1 |
| 02/03/2019 | 08/10/2021 | 15/03/2019 | R10B | R8 | | | | | | 35 |
| 02/03/2019 | 02/10/2019 | 08/10/2021 | R10B | R8 | | | | | | 1 |
| 02/10/2019 | 08/10/2021 | 17/07/2020 | R10B | R8 | | | | | | 6 |
| 03/11/2020 | 08/10/2021 | 17/07/2020 | R10B | R8 | | | | | | 1 |

| Intersection between | | | | | | | | | | #MF |
|---|---|---|---|---|---|---|---|---|---|---|
| 02/03/2019 | 02/10/2019 | 08/10/2021 | 15/03/2019 | R8 | | | | | | 8 |
| 02/03/2019 | 03/11/2020 | 08/10/2021 | 15/03/2019 | R8 | | | | | | 1 |
| 02/10/2019 | 03/11/2020 | 08/10/2021 | 17/07/2020 | R8 | | | | | | 1 |
| 02/03/2019 | 02/10/2019 | 15/03/2019 | R10B | R9 | | | | | | 1 |
| 02/03/2019 | 08/10/2021 | 15/03/2019 | R10B | R9 | | | | | | 28 |
| 02/03/2019 | 02/10/2019 | 08/10/2021 | R10B | R9 | | | | | | 1 |
| 02/10/2019 | 08/10/2021 | 15/03/2019 | R10B | R9 | | | | | | 1 |
| 02/10/2019 | 08/10/2021 | 17/07/2020 | R10B | R9 | | | | | | 2 |
| 02/03/2019 | 02/10/2019 | 08/10/2021 | 15/03/2019 | R9 | | | | | | 2 |
| 02/03/2019 | 02/10/2019 | 08/10/2021 | 15/03/2019 | R10B | | | | | | 2 |
| 02/03/2019 | 02/10/2019 | 08/10/2021 | 17/07/2020 | R10B | | | | | | 1 |
| 02/10/2019 | 03/11/2020 | 08/10/2021 | 17/07/2020 | R10B | | | | | | 2 |
| 02/03/2019 | 02/10/2019 | 08/10/2021 | 15/03/2019 | 22/10/2019 | | | | | | 1 |
| 02/03/2019 | 02/10/2019 | 03/11/2020 | 15/03/2019 | 17/07/2020 | | | | | | 1 |
| 02/03/2019 | 02/10/2019 | 08/10/2021 | 15/03/2019 | 17/07/2020 | | | | | | 30 |
| 02/03/2019 | 02/10/2019 | 03/11/2020 | 08/10/2021 | 15/03/2019 | | | | | | 13 |
| 02/03/2019 | 02/10/2019 | 03/11/2020 | 08/10/2021 | 17/07/2020 | | | | | | 1 |
| 02/10/2019 | 03/11/2020 | 15/03/2019 | 17/07/2020 | 22/10/2019 | | | | | | 1 |
| 02/10/2019 | 03/11/2020 | 08/10/2021 | 15/03/2019 | 17/07/2020 | | | | | | 6 |
| 02/10/2019 | 03/11/2020 | 08/10/2021 | 17/07/2020 | 22/10/2019 | | | | | | 6 |
| 02/03/2019 | R10B | R8 | R9 | | | | | | | 64 |
| 15/03/2019 | R10B | R8 | R9 | | | | | | | 15 |
| 02/10/2019 | R10B | R8 | R9 | | | | | | | 50 |
| 03/11/2020 | R10B | R8 | R9 | | | | | | | 1 |
| 08/10/2021 | R10B | R8 | R9 | | | | | | | 135 |
| 02/03/2019 | 15/03/2019 | R8 | R9 | | | | | | | 7 |
| 02/03/2019 | 02/10/2019 | R8 | R9 | | | | | | | 1 |
| 02/03/2019 | 08/10/2021 | R8 | R9 | | | | | | | 71 |
| 08/10/2021 | 15/03/2019 | R8 | R9 | | | | | | | 7 |
| 02/10/2019 | 08/10/2021 | R8 | R9 | | | | | | | 24 |
| 08/10/2021 | 17/07/2020 | R8 | R9 | | | | | | | 1 |
| 02/03/2019 | 15/03/2019 | R10B | R8 | | | | | | | 4 |
| 02/03/2019 | 08/10/2021 | R10B | R8 | | | | | | | 14 |
| 08/10/2021 | 15/03/2019 | R10B | R8 | | | | | | | 5 |
| 02/10/2019 | 08/10/2021 | R10B | R8 | | | | | | | 4 |
| 08/10/2021 | 17/07/2020 | R10B | R8 | | | | | | | 3 |
| 02/03/2019 | 02/10/2019 | 15/03/2019 | R8 | | | | | | | 2 |
| 02/03/2019 | 08/10/2021 | 15/03/2019 | R8 | | | | | | | 53 |
| 02/03/2019 | 02/10/2019 | 08/10/2021 | R8 | | | | | | | 4 |

| Intersection between | | | | | | | | | | #MF |
|---|---|---|---|---|---|---|---|---|---|---|
| 02/03/2019 | 03/11/2020 | 08/10/2021 | R8 | | | | | | | 1 |
| 02/10/2019 | 08/10/2021 | 15/03/2019 | R8 | | | | | | | 1 |
| 02/10/2019 | 08/10/2021 | 22/10/2019 | R8 | | | | | | | 2 |
| 02/10/2019 | 03/11/2020 | 17/07/2020 | R8 | | | | | | | 1 |
| 02/10/2019 | 08/10/2021 | 17/07/2020 | R8 | | | | | | | 10 |
| 03/11/2020 | 08/10/2021 | 17/07/2020 | R8 | | | | | | | 2 |
| 02/03/2019 | 15/03/2019 | R10B | R9 | | | | | | | 20 |
| 02/03/2019 | 02/10/2019 | R10B | R9 | | | | | | | 2 |
| 02/03/2019 | 08/10/2021 | R10B | R9 | | | | | | | 24 |
| 08/10/2021 | 15/03/2019 | R10B | R9 | | | | | | | 2 |
| 02/10/2019 | 08/10/2021 | R10B | R9 | | | | | | | 2 |
| 02/03/2019 | 08/10/2021 | 15/03/2019 | R9 | | | | | | | 23 |
| 02/03/2019 | 02/10/2019 | 08/10/2021 | R9 | | | | | | | 2 |
| 02/10/2019 | 03/11/2020 | 17/07/2020 | R9 | | | | | | | 1 |
| 02/03/2019 | 08/10/2021 | 15/03/2019 | R10B | | | | | | | 16 |
| 02/10/2019 | 08/10/2021 | 17/07/2020 | R10B | | | | | | | 2 |
| 02/03/2019 | 02/10/2019 | 15/03/2019 | 17/07/2020 | | | | | | | 2 |
| 02/03/2019 | 02/10/2019 | 08/10/2021 | 15/03/2019 | | | | | | | 60 |
| 02/03/2019 | 08/10/2021 | 15/03/2019 | 22/10/2019 | | | | | | | 1 |
| 02/03/2019 | 08/10/2021 | 15/03/2019 | 17/07/2020 | | | | | | | 6 |
| 02/03/2019 | 03/11/2020 | 08/10/2021 | 15/03/2019 | | | | | | | 10 |
| 02/03/2019 | 02/10/2019 | 17/07/2020 | 22/10/2019 | | | | | | | 1 |
| 02/03/2019 | 02/10/2019 | 03/11/2020 | 17/07/2020 | | | | | | | 1 |
| 02/03/2019 | 02/10/2019 | 08/10/2021 | 17/07/2020 | | | | | | | 2 |
| 02/03/2019 | 03/11/2020 | 08/10/2021 | 17/07/2020 | | | | | | | 1 |
| 02/10/2019 | 08/10/2021 | 15/03/2019 | 17/07/2020 | | | | | | | 2 |
| 02/10/2019 | 03/11/2020 | 17/07/2020 | 22/10/2019 | | | | | | | 3 |
| 02/10/2019 | 08/10/2021 | 17/07/2020 | 22/10/2019 | | | | | | | 1 |
| 02/10/2019 | 03/11/2020 | 08/10/2021 | 17/07/2020 | | | | | | | 32 |
| R10B | R8 | R9 | | | | | | | | 123 |
| 02/03/2019 | R8 | R9 | | | | | | | | 45 |
| 15/03/2019 | R8 | R9 | | | | | | | | 5 |
| 02/10/2019 | R8 | R9 | | | | | | | | 5 |
| 08/10/2021 | R8 | R9 | | | | | | | | 147 |
| 02/03/2019 | R10B | R8 | | | | | | | | 5 |
| 15/03/2019 | R10B | R8 | | | | | | | | 3 |
| 17/07/2020 | R10B | R8 | | | | | | | | 1 |
| 08/10/2021 | R10B | R8 | | | | | | | | 52 |
| 02/03/2019 | 15/03/2019 | R8 | | | | | | | | 10 |

| Intersection between | | | | | | | | | | #MF |
|---|---|---|---|---|---|---|---|---|---|---|
| 02/03/2019 | 08/10/2021 | R8 | | | | | | | | 65 |
| 08/10/2021 | 15/03/2019 | R8 | | | | | | | | 18 |
| 02/10/2019 | 17/07/2020 | R8 | | | | | | | | 5 |
| 02/10/2019 | 08/10/2021 | R8 | | | | | | | | 57 |
| 08/10/2021 | 17/07/2020 | R8 | | | | | | | | 6 |
| 03/11/2020 | 08/10/2021 | R8 | | | | | | | | 1 |
| 02/03/2019 | R10B | R9 | | | | | | | | 44 |
| 15/03/2019 | R10B | R9 | | | | | | | | 2 |
| 02/10/2019 | R10B | R9 | | | | | | | | 3 |
| 17/07/2020 | R10B | R9 | | | | | | | | 1 |
| 08/10/2021 | R10B | R9 | | | | | | | | 24 |
| 02/03/2019 | 15/03/2019 | R9 | | | | | | | | 9 |
| 02/03/2019 | 02/10/2019 | R9 | | | | | | | | 1 |
| 02/03/2019 | 08/10/2021 | R9 | | | | | | | | 21 |
| 08/10/2021 | 15/03/2019 | R9 | | | | | | | | 6 |
| 02/10/2019 | 08/10/2021 | R9 | | | | | | | | 3 |
| 08/10/2021 | 17/07/2020 | R9 | | | | | | | | 1 |
| 02/03/2019 | 15/03/2019 | R10B | | | | | | | | 9 |
| 02/03/2019 | 02/10/2019 | R10B | | | | | | | | 2 |
| 02/03/2019 | 08/10/2021 | R10B | | | | | | | | 10 |
| 08/10/2021 | 15/03/2019 | R10B | | | | | | | | 5 |
| 02/10/2019 | 08/10/2021 | R10B | | | | | | | | 4 |
| 02/03/2019 | 02/10/2019 | 15/03/2019 | | | | | | | | 3 |
| 02/03/2019 | 03/11/2020 | 15/03/2019 | | | | | | | | 2 |
| 02/03/2019 | 08/10/2021 | 15/03/2019 | | | | | | | | 203 |
| 02/03/2019 | 02/10/2019 | 08/10/2021 | | | | | | | | 6 |
| 02/03/2019 | 03/11/2020 | 17/07/2020 | | | | | | | | 1 |
| 02/03/2019 | 08/10/2021 | 17/07/2020 | | | | | | | | 2 |
| 02/10/2019 | 03/11/2020 | 15/03/2019 | | | | | | | | 1 |
| 02/10/2019 | 08/10/2021 | 15/03/2019 | | | | | | | | 9 |
| 08/10/2021 | 15/03/2019 | 17/07/2020 | | | | | | | | 1 |
| 03/11/2020 | 08/10/2021 | 15/03/2019 | | | | | | | | 1 |
| 02/10/2019 | 17/07/2020 | 22/10/2019 | | | | | | | | 7 |
| 02/10/2019 | 03/11/2020 | 22/10/2019 | | | | | | | | 1 |
| 02/10/2019 | 08/10/2021 | 22/10/2019 | | | | | | | | 2 |
| 02/10/2019 | 03/11/2020 | 17/07/2020 | | | | | | | | 11 |
| 02/10/2019 | 08/10/2021 | 17/07/2020 | | | | | | | | 51 |
| 02/10/2019 | 03/11/2020 | 08/10/2021 | | | | | | | | 1 |
| 03/11/2020 | 08/10/2021 | 17/07/2020 | | | | | | | | 3 |

| Intersection between | | | | | | | | | | #MF |
|---|---|---|---|---|---|---|---|---|---|---|
| R8 | R9 | | | | | | | | | 68 |
| R10B | R8 | | | | | | | | | 49 |
| 02/03/2019 | R8 | | | | | | | | | 31 |
| 15/03/2019 | R8 | | | | | | | | | 9 |
| 02/10/2019 | R8 | | | | | | | | | 2 |
| 17/07/2020 | R8 | | | | | | | | | 1 |
| 08/10/2021 | R8 | | | | | | | | | 370 |
| R10B | R9 | | | | | | | | | 70 |
| 02/03/2019 | R9 | | | | | | | | | 26 |
| 15/03/2019 | R9 | | | | | | | | | 4 |
| 17/07/2020 | R9 | | | | | | | | | 1 |
| 08/10/2021 | R9 | | | | | | | | | 34 |
| 02/03/2019 | R10B | | | | | | | | | 27 |
| 15/03/2019 | R10B | | | | | | | | | 9 |
| 02/10/2019 | R10B | | | | | | | | | 2 |
| 08/10/2021 | R10B | | | | | | | | | 22 |
| 02/03/2019 | 15/03/2019 | | | | | | | | | 131 |
| 02/03/2019 | 02/10/2019 | | | | | | | | | 1 |
| 02/03/2019 | 17/07/2020 | | | | | | | | | 1 |
| 02/03/2019 | 08/10/2021 | | | | | | | | | 245 |
| 02/10/2019 | 15/03/2019 | | | | | | | | | 2 |
| 08/10/2021 | 15/03/2019 | | | | | | | | | 67 |
| 02/10/2019 | 22/10/2019 | | | | | | | | | 1 |
| 02/10/2019 | 17/07/2020 | | | | | | | | | 57 |
| 02/10/2019 | 03/11/2020 | | | | | | | | | 2 |
| 02/10/2019 | 08/10/2021 | | | | | | | | | 91 |
| 03/11/2020 | 22/10/2019 | | | | | | | | | 1 |
| 08/10/2021 | 22/10/2019 | | | | | | | | | 1 |
| 03/11/2020 | 17/07/2020 | | | | | | | | | 2 |
| 08/10/2021 | 17/07/2020 | | | | | | | | | 26 |
| 03/11/2020 | 08/10/2021 | | | | | | | | | 5 |
| *R8* | | | | | | | | | | *233* |
| *R9* | | | | | | | | | | *123* |
| *R10B* | | | | | | | | | | *209* |
| *02/03/2019* | | | | | | | | | | *626* |
| *15/03/2019* | | | | | | | | | | *134* |
| *02/10/2019* | | | | | | | | | | *290* |
| *22/10/2019* | | | | | | | | | | *7* |
| *17/07/2020* | | | | | | | | | | *78* |

| Intersection between | | | | | | | | | #MF |
|---|---|---|---|---|---|---|---|---|---|
| *03/11/2020* | | | | | | | | | *11* |
| *08/10/2021* | | | | | | | | | *3929* |

We also calculated the intersections of the lists of MFs, which are presented in the following table, where samples are reported in blue, the total number of MFs is reported next to the sample name in cyan. Lines in green shades reports the number of MFs in common between two samples. In the line below, in red and blue shades, are reported the percentage of MFs in common on the total number of MFs for each samples. For instance, for the intersection between R10B and R9, 1754 MFs are in common, which represent 77% of the number of MFs in R10B and 70% of the MFs in R9. The higher the two percentages, the higher the similarity of the MFs in the two samples compared. To increase the readability of the table, higher percentages are in red and blue, while low percentages are faded.

At first glance we can see the high similarity of the composition of R8, R9 and R10B, which contain MFs that are also present in the sample 08/10/2021. Samples 02/03/2019 and 15/03/2019 have a quite similar composition and contain more than 1000 MFs detected also in R8, R, R9 and R10B. Nevertheless, this similarity is less visible between the other samples collected at PUY and R10B. Interestingly, even if samples collected at PUY in autumn and summer contain a lower number of MFs, the composition is similar.

In summary, it seems that MFs contained in samples collected during summer and autumn at PUY, with the exception pf 08/10/2021, are different from those observed in samples collected at REU and could justify the different OSC value. Nevertheless, samples collected in winter at PUY are relatively similar to those collected in REU, although the OSC value is significantly different between these samples. Sample 08/10/2021 has such a large variety of MFs that is similar to both samples collected at REU and at PUY.

|  |  | R9 | R10B | 02/03/2019 | 15/03/2019 | 02/10/2019 | 22/10/2019 | 17/07/2020 | 03/11/2020 | 08/10/2021 |
|---|---|---|---|---|---|---|---|---|---|---|
|  |  | 2503 | 2276 | 3244 | 2084 | 1543 | 120 | 715 | 312 | 7436 |
| **R8** | 3098 | 2001 | 1708 | 1517 | 1151 | 701 | 34 | 279 | 108 | 2274 |
|  | % ↑ | 80% | 75% | 47% | 55% | 45% | 28% | 39% | 35% | 31% |
|  | % ← | 65% | 55% | 49% | 37% | 23% | 1% | 9% | 3% | 73% |
| **R9** | 2503 |  | 1754 | 1488 | 1104 | 617 | 32 | 241 | 95 | 1728 |
|  | % ↑ |  | 77% | 46% | 53% | 40% | 27% | 34% | 30% | 23% |
|  | % ← |  | 70% | 59% | 44% | 25% | 1% | 10% | 4% | 69% |
| **R10B** | 2276 |  |  | 1322 | 1049 | 584 | 29 | 246 | 96 | 1477 |
|  | % ↑ |  |  | 41% | 50% | 38% | 24% | 34% | 31% | 20% |
|  | % ← |  |  | 58% | 46% | 26% | 1% | 11% | 4% | 65% |
| 02/03/2019 | 3244 |  |  |  | 1722 | 648 | 84 | 295 | 192 | 2085 |
|  | % ↑ |  |  |  | 83% | 42% | 70% | 41% | 62% | 28% |
|  | % ← |  |  |  | 53% | 20% | 3% | 9% | 6% | 64% |
| 15/03/2019 | 2084 |  |  |  |  | 620 | 85 | 295 | 198 | 1620 |
|  | % ↑ |  |  |  |  | 40% | 71% | 41% | 63% | 22% |
|  | % ← |  |  |  |  | 30% | 4% | 14% | 10% | 78% |
| 02/10/2019 | 1543 |  |  |  |  |  | 107 | 558 | 260 | 1069 |
|  | % ↑ |  |  |  |  |  | 89% | 78% | 83% | 14% |
|  | % ← |  |  |  |  |  | 7% | 36% | 17% | 69% |
| 22/10/2019 | 120 |  |  |  |  |  |  | 87 | 68 | 98 |
|  | % ↑ |  |  |  |  |  |  | 12% | 22% | 1% |
|  | % ← |  |  |  |  |  |  | 73% | 57% | 82% |
| 17/07/2020 | 715 |  |  |  |  |  |  |  | 227 | 531 |
|  | % ↑ |  |  |  |  |  |  |  | 73% | 7% |
|  | % ← |  |  |  |  |  |  |  | 32% | 74% |
| 03/11/2020 | 312 |  |  |  |  |  |  |  |  | 266 |
|  | % ↑ |  |  |  |  |  |  |  |  | 4% |
|  | % ← |  |  |  |  |  |  |  |  | 85% |

Lines 362-364: If I am understanding correctly, the general percentage of formulas in each classification is similar between REU and PUY, which seems reasonable, I am still curious about the specific differences

between the molecules in one sample or another in a more comprehensive view. Do the formulas in each classification match each other between the different sites or are they largely different? For example, for the LipidC classification, are the formulas found at REU and PUY 90% common, 70%, 50%, less? I think it could be interesting to see if the detailed composition of these samples is very different or the same, since it may say something about the cloud processing results. The "averages" are very useful, but as you have mentioned, even the same formula doesn't necessarily mean the same molecule, so if a set of molecular formulas are in a particular classification, they may not be similar in any other way, or they could be very similar and highlight that cloud processing brings organic matter to a similar specific result.

We thank the reviewer for this useful comment. Indeed, results are quite surprising for the specific dataset. The comparison of lipids MFs in REU samples highlighted that 48% of the MFs are common to the three samples. Intuitively, we could imagine a similar result for PUY samples: nevertheless, the comparison of lipids MFs shows that only 3% of the total number of lipid MFs (1303) is in common between all the samples. The percentage reaches 4.1% for common MFs in 6 (out of 7) samples, 6.2% for common MFs in 5 samples and 10.3% for common MFs in 4 samples. The comparison between samples collected in PUY and REU is reported in the following table, which shows the number of lipids MFs in each samples (for instance, 4 MFs are present in all ten samples, the total number of lipid MFs (1903) and the % of MFs on the total number. Only 0.2% of the MFs are common to all the samples, showing that lipids from REU are different from lipids from PUY samples.

| occurrence in samples | 10 | 9 | 8 | 7 | 6 | 5 | 4 | 3 | 2 |
|---|---|---|---|---|---|---|---|---|---|
| # of MFs | 4 | 28 | 60 | 120 | 356 | 259 | 315 | 379 | 382 |
| total # of MFs (lipids) | 1903 | | | | | | | | |
| % | 0.2 | 1.5 | 3.2 | 6.3 | 18.7 | 13.6 | 16.6 | 19.9 | 20.1 |

Considering these results, we could argue that cloud processing increases the molecular complexity of cloud water, and we need to look at the final products of cloud processing, such as C1-C3 compounds to find similarity.

A sentence is added in the article.

Lines 386-387: "Although LipidC represents more than half of the MFs in REU and PUY samples, only four MFs of this class are in common between all the samples."

Lines 370: While the FT-ICR is very well suited and effective for this work, the lack of structural information is a shortcoming as noted here, is there any interest in doing LC or fragmentation analysis in the future for these samples or others?

We agree with the reviewer: we need more information on the structure. Nevertheless, it is still difficult to combine LC or fragmentation analysis with FT-ICR MS for complex matrices. A nice work from Divisekara et al., (2023) reports the development of software for this kind of analysis and we would be glad to have the opportunity to test this approach on our samples.

Lines 378-384: You are taking appropriate caution in classifying these molecules as one specific class or another with the database, but I was curious whether if you took a few of the formulas that you have classified as "prenol lipids" for example and just looked for any molecule matching that formula (in other databases or the search engine of your choice) if you could get any other classification?

A quick research of some molecular formula in Chemspider (https://www.chemspider.com/) confirm that we need to interpret our results with extreme caution. For instance, the MF corresponding to myrtenic acid $C_{10}H_{14}O_2$ produced 5609 results in Chemspider and myrtenic is the 80[th] of the list. Similarly, the MF corresponding to parthenin, $C_{15}H_{18}O_4$, gave 7717 results, with parthenin in 27[th] position. That's the reason of our caution in this interpretation and why we will be glad to analyze samples in LC-FT-ICR MS.

Lines 426-428: I do not quite understand this sentence. Are the measured concentrations for alpha pinene 0.5, 71.5, and 2 for R8, R9, and R10B, while the beta pinene concentrations were 39.9 and 1.3 for R8, R9, and R10B, or are the detection limits for alpha pinene 39.9 and for beta pinene they are 1.3? I think the sentence could be restructured for clarity.

We agree with the reviewer. The sentence was modified as follows

Lines 451-453: "In particular, alpha pinene concentrations were 0.5, 71.5 and 2.0 nmol L$^{-1}$ and beta pinene concentrations were below detection limit for R8 and 39.9 and 1.3 nmol L$^{-1}$ for R9 and R10B, respectively."

Lines 457: Does this mean that the organosulfate intensity was low in all samples (REU and PUY) with the exception of PUY 8/10/2021, or are you just comparing PUY 08/10/2021 to other PUY samples? Additionally, you explain the higher occurrence of limonene organosulfates at REU by the increased emission of limonene at the site, which makes sense, but does that imply that the organosulfate formation from limonene is a faster process than the oxidation of pinene? My understanding of the reason given for the relative lack of pinene oxidation products is that the emissions were too fresh to have oxidized yet. Is the organosulfate a primary oxidation product like C8H12O5? Or is the explanation that there is more limonene emissions relative to the pinenes?

The organosulfate intensity is higher in the 8/10/2021 sample than in the other samples from the PUY and the REU. The text was modified to clarify the statement. At the REU, limonene emissions are mostly on the coastal region while pinene emissions are prevalent on the slope of the mountain, where samples are collected. That implies that pinene emissions are fresher than limonene emissions. That is the reason why limonene is more oxidized than pinene. We think that the formation of organosulfates in the coastal environment is favored due to the strong emission of NOx from traffic and DMS from the sea, as evidenced in the work from Rocco et al., (2022).

Lines 465: What were the N and S beta caryophalene formulas? Is there any way to know that the formulas are N or S caryophyllene molecules other than matching the formulas? While presence of their emission sources on the coast may explain the N and S beta caryophyllene, why would there be no CHO oxidation products? Are the N and S reactions that much more favorable than the O oxidation? Or is the concentration of N and S so overwhelming that the O oxidation doesn't really occur, relative to N and S?

The hypothesis presented is just a speculation and cannot be supported by the dataset. Thus, it was removed from the text.

Line 524: According to the classification you say that 50% of the molecules observed are reduced, is the explanation that the organic matter in the clouds is fairly fresh and hasn't had a chance to oxidize more completely yet?

This comment is interesting and the hypothesis mentioned could explain the presence of lipids in REU samples. Nevertheless, local sources and an input of fresh organic matter can be excluded at PUY. One

potential explanation is that reduced organic matter, which is likely to be hydrophobic or amphiphilic, such as fatty acids, is located at the interface between water and air and reacts less with photogenerated radicals in the aqueous phase. This can preserve it from the oxidation. When we collect the sample by collision of the droplets on the plates of the collector and coalescence into the bottle, the reduced matter is trapped into the liquid and then, concentrated by SPE and analyzed by FT-ICR MS. This is just a hypothesis that cannot be proved by our results, thus no modifications are reported in the article.

Technical Corrections

Line 25: Somewhat contradictory statements, can consider changing the language a bit to get to the assumed intended meaning.

We agree with the reviewer. The text was modified as follow:

Lines 24-28: "The composition of cloud water dissolved organic matter has been investigated through non-targeted high resolution mass spectrometry only on few samples that were mostly collected in the Northern hemisphere, in USA, Europe and China. There remains, therefore, a lack of measurements for clouds located in the Southern Hemisphere, under tropical conditions and influenced by forest emissions. As a matter of fact, the comparison of the composition of clouds collected in different locations is challenging since the methodology for the analysis and data treatment are not standardized."

Line 179: It may be more consistent and precise to say "same mass" instead of "same peak", since the parenthetical on line 180 says "unique mass".

We agree with the reviewer. The text was modified as follow:

Lines 179-181: The output of the function gives a list of ambiguous (multiple MFs that have been assigned to the same mass) and unambiguous (MFs that have been assigned to a unique mass) MFs.

Lines 323: Should probably change "is" to "are"

We agree with the reviewer. The text was corrected

Lines 520: Instead of "emitted" you should probably say something like "developed" or "produced". Overall the language in this manuscript is very good, but there are few minor things, like this and the comments for lines 426-428 that could be adjusted.

We agree with the reviewer. The text was modified as follow:

Lines 546-548: "We hypothesized that, for autumn samples, strong emissions are rapidly processed at REU, due to the high temperature, and aged air masses are collected at PUY, leading to similar values of average OSC, but produced by different causes."

**References**

Bianco, A., Deguillaume, L., Vaïtilingom, M., Nicol, E., Baray, J.-L., Chaumerliac, N., and Bridoux, M. C.: Molecular Characterization of Cloud Water Samples Collected at the puy de Dôme (France) by Fourier Transform Ion Cyclotron Resonance Mass Spectrometry, Environmental Science & Technology, https://doi.org/10.1021/acs.est.8b01964, 2018.

Bianco, A., Riva, M., Baray, J.-L., Ribeiro, M., Chaumerliac, N., George, C., Bridoux, M., and Deguillaume, L.: Chemical Characterization of Cloudwater Collected at Puy de Dôme by FT-ICR MS Reveals the Presence of SOA Components, ACS Earth Space Chem., 3, 2076–2087, https://doi.org/10.1021/acsearthspacechem.9b00153, 2019.

Cook, R. D., Lin, Y.-H., Peng, Z., Boone, E., Chu, R. K., Dukett, J. E., Gunsch, M. J., Zhang, W., Tolic, N., and Laskin, A.: Biogenic, urban, and wildfire influences on the molecular composition of dissolved organic compounds in cloud water, Atmospheric Chemistry and Physics, 17, 15167–15180, 2017.

Divisekara, T., Schum, S., and Mazzoleni, L.: Ultrahigh performance LC/FT-MS non-targeted screening for biomass burning organic aerosol with MZmine2 and MFAssignR, Chemosphere, 338, 139403, https://doi.org/10.1016/j.chemosphere.2023.139403, 2023.

Rocco, M., Baray, J. -L., Colomb, A., Borbon, A., Dominutti, P., Tulet, P., Amelynck, C., Schoon, N., Verreyken, B., Duflot, V., Gros, V., Sarda-Estève, R., Péris, G., Guadagno, C., and Leriche, M.: High Resolution Dynamical Analysis of Volatile Organic Compounds (VOC) Measurements During the BIO-MAÏDO Field Campaign (Réunion Island, Indian Ocean), JGR Atmospheres, 127, https://doi.org/10.1029/2021JD035570, 2022.

Sun, W., Fu, Y., Zhang, G., Yang, Y., Jiang, F., Lian, X., Jiang, B., Liao, Y., Bi, X., Chen, D., Chen, J., Wang, X., Ou, J., Peng, P., and Sheng, G.: Measurement report: Molecular characteristics of cloud water in southern China and insights into aqueous-phase processes from Fourier transform ion cyclotron resonance mass spectrometry, Atmos. Chem. Phys., 21, 16631–16644, https://doi.org/10.5194/acp-21-16631-2021, 2021.

Zhao, Y., Hallar, A. G., and Mazzoleni, L. R.: Atmospheric organic matter in clouds: exact masses and molecular formula identification using ultrahigh-resolution FT-ICR mass spectrometry, Atmospheric Chemistry and Physics, 13, 12343–12362, 2013.